# Transcriptional landscapes underlying Notch-induced lineage conversion and plasticity of mammary basal cells

Candice Merle [1], Calvin Rodrigues [1], Atefeh Pourkhalili Langeroudi[1], Robin Journot[1], Fabian Rost [2,3], Yiteng Dang[2,3,4], Steffen Rulands [5] & Silvia Fre [1]✉

## Abstract

The mammary epithelium derives from multipotent mammary stem cells (MaSCs) that engage into differentiation during embryonic development. However, adult MaSCs maintain the ability to reactivate multipotency in non-physiological contexts. We previously reported that Notch1 activation in committed basal cells triggers a basal-to-luminal cell fate switch in the mouse mammary gland. Here, we report conservation of this mechanism and found that in addition to the mammary gland, constitutive Notch1 signaling induces a basal-to-luminal cell fate switch in adult cells of the lacrimal gland, the salivary gland, and the prostate. Since the lineage transition is progressive in time, we performed single-cell transcriptomic analysis on index-sorted mammary cells at different stages of lineage conversion, generating a temporal map of changes in cell identity. Combining single-cell analyses with organoid assays, we demonstrate that cell proliferation is indispensable for this lineage conversion. We also reveal the individual transcriptional landscapes underlying the cellular plasticity switching of committed mammary cells in vivo with spatial and temporal resolution. Given the roles of Notch signaling in cancer, these results may help to better understand the mechanisms that drive cellular transformation.

**Key words** Lineage Conversion; Notch1 Signaling; Epithelial Stem Cells; Plasticity
**Subject Categories** Chromatin, Transcription & Genomics; Development; Stem Cells & Regenerative Medicine

## Introduction

The adult mammary gland (MG) is composed of two epithelial layers featuring two main cell types: basal cells (BCs) in contact with the basement membrane and luminal cells (LCs) facing the ductal lumen, which can be further subdivided in ERα$^{pos}$/PR$^{pos}$ (HR$^{pos}$) and ERα$^{neg}$/PR$^{neg}$ (HR$^{neg}$) cells. It is now well-established that this tissue is maintained by unipotent lineage-restricted progenitors throughout adult life in homeostatic conditions, but these self-renewing committed cells retain a high degree of plasticity, as they can revert to multipotency in several circumstances.

This was first observed in transplantation experiments, when adult unipotent mammary cells displayed multilineage differentiation capacity and could generate a functional mammary gland composed of both BCs and LCs (Rodilla et al, 2015; Shackleton et al, 2006; Stingl et al, 2006; Van Keymeulen et al, 2011). Other experimental procedures involving the dissociation of BCs and LCs, including when they are grown separately in 3D organoid conditions, have shown reactivation of multipotency of BCs, that can also be induced by different types of epithelial injury causing tissue regeneration and, importantly, by oncogene activation (Jamieson et al, 2017; Jardé et al, 2016; Koren et al, 2015; Van Keymeulen et al, 2015). Moreover, recent elegant in vivo experiments illustrated the capacity of BCs to generate LCs upon genotoxic stress (Seldin and Macara, 2020) or LCs genetic ablation (Centonze et al, 2020), strongly suggesting that LCs restrict the default multipotency of BCs. We have previously shown that the binary fate choice between basal or luminal commitment is controlled by Notch signaling, a master regulator of cell fate choices in most vertebrate and invertebrate tissues, which is both necessary and sufficient for luminal fate specification in the mammary gland. Importantly, our previous studies uncovered that, besides its essential role in controlling fate decisions of embryonic multipotent mammary stem cells, constitutive and ectopic Notch activation in committed adult BCs, which never experience Notch activity, can also "reprogram" their lineage potential and induce their conversion into ERα$^{neg}$/PR$^{neg}$ luminal cells (Lilja et al, 2018).

The "reprogramming" capacity of mammary progenitors has important implications for cell differentiation as well as transformation; given that the cell fate switch did not occur in all cells at the same time, we set out to assess if the targeted BCs themselves transdifferentiate into LCs or if they respond to Notch activation by giving rise to luminal daughter cells. To this aim, we investigated the

[1]Institut Curie, Laboratory of Genetics and Developmental Biology, INSERM U934, CNRS UMR3215, PSL University, Sorbonne University, Paris, France. [2]Max-Planck-Institute for the Physics of Complex Systems, Dresden, Germany. [3]Max Planck Institute for Molecular Cell Biology and Genetics, Dresden, Germany. [4]Center for Systems Biology, Dresden, Germany. [5]Ludwig-Maximilians-Universität München, Arnold-Sommerfeld-Center for Theoretical Physics, München, Germany. ✉E-mail: silvia.fre@curie.fr

dynamics of the progressive lineage transition from basal to luminal fate to understand how the Notch-imposed cell fate switch is mechanistically achieved and to reveal the changes in transcriptional state of single Notch mutant cells in vivo, using two different Cre promoters and single-cell RNA sequencing of index-sorted mutant cells at different stages of lineage transition. We found that the transcriptional changes associated with the transition from basal to luminal fate are progressive in time, triggered by the initial decrease in basal markers expression followed by the steady upregulation of luminal genes. Thanks to organoid cultures, we could also establish that proliferation is essential for cell fate conversion to occur, ruling out the possibility of a transdifferentiation event.

# Results

## Ectopic Notch activation induces a cell fate switch in four different bi-layered epithelia

We have previously shown that constitutive activation of Notch signaling through the ectopic expression of the ligand-independent, intracellular portion of the mouse Notch1 receptor (in R26-N1ICD-ires-nGFP gain-of-function mice) (Murtaugh et al, 2003) in mammary BCs, targeted by two different BC-specific inducible Cre promoters, SMACre$^{ERT2}$ and K5Cre$^{ERT2}$, is sufficient to trigger a progressive switch in cell fate and eventually forces all mutant cells to acquire a luminal hormone receptor$^{neg}$ (HR$^{neg}$) identity (Lilja et al, 2018).

Importantly, the cell fate switch from BCs to LCs happens progressively over a long period of time (6 weeks) (Fig. 1A–C). At mid-transition (3 weeks upon induction), cells featuring nuclear GFP (nGFP$^{pos}$), reporting Notch pathway activation, co-express basal (α-SMA or K14) and luminal (K8) markers (Fig. 1A,B), reminiscent of embryonic multipotent MaSCs. This cell fate transition happens in unipotent adult BCs, as demonstrated by the exclusive labeling of α-SMA$^{pos}$ or K5$^{pos}$ BCs in control SMACre$^{ERT2}$/mTmG or K5Cre$^{ERT2}$/mTmG mice, tracing the fate of targeted BCs and their progeny (Fig. EV1A).

Throughout the lineage switch, hybrid cells co-expressing luminal and basal markers can be scored by flow cytometry analysis as dispersed cells laying between the luminal (EPCAM$^{high}$/CD49f$^{low}$) and the basal (EPCAM$^{low}$/CD49$^{high}$) gates (Fig. EV1B,C). The proportion of cells transiting through this intermediate state is highest mid-way through the lineage transition, around 3–4 weeks after Notch activation (Figs. 1C and EV1C). Eventually, the intermediate population is no longer detected after a 6-week chase, and the vast majority of mutant nGFP$^{pos}$ cells have become luminal (Fig. 1C).

The Notch pathway is crucial for stem cell fate decisions in a variety of tissues. To assess the conservation of the role of Notch in binary cell fate choices in other glandular epithelia, we then analyzed the plasticity and differentiation state of unipotent adult BCs induced to express N1ICD by the SMACre promoter (SMACre$^{ERT2}$/N1ICD-nGFP) in the salivary gland (SG) and the lacrimal gland (LG), and by the K5Cre promoter (K5Cre$^{ERT2}$/N1ICD-nGFP) in the prostate. Showcasing remarkable conservation, N1ICD-expressing cells acquired a luminal identity in the SG, LG, and prostate within the same temporal window (6-week chase) (Fig. 1D–G), whereas control mGFP$^{pos}$ BCs maintained their basal

identity in both SMACre$^{ERT2}$/mTmG mice or K5Cre$^{ERT2}$/mTmG mice in all examined tissues (Fig. EV1A).

The striking conservation of the phenotype induced by ectopic Notch activation in four adult epithelia derived from different germ layers highlights the essential role of Notch signaling as a broad determinant of luminal cell fate.

## Intermediate mammary cells feature a hybrid transcriptional signature

The conspicuous robustness of the results we obtained using the same gain-of-function mutant mice in four different adult tissues demonstrates the high degree of plasticity of lineage-committed unipotent basal progenitors, that can readily change fate if homeostasis is perturbed, as previously observed in organoids (Jamieson et al, 2017; Jardé et al, 2016), in transplantation assays (Van Keymeulen et al, 2011) or, more recently, upon genotoxic agents exposure (Seldin and Macara, 2020) and in genetic ablation experiments in vivo (Centonze et al, 2020).

To gain mechanistic insights into the observed Notch-imposed cell fate switch and reveal the molecular pathways involved, we set out to decipher the properties of individual intermediate mammary cells at the single-cell level, to capture discrete gene expression states that could represent distinct differentiation trajectories. To this aim, we performed single-cell RNA sequencing (scRNAseq) by Smart-seq2 on index-sorted mammary BCs, intermediate cells and LCs, both GFP$^{neg}$ and GFP$^{pos}$, isolated from mammary glands of SMACre$^{ERT2}$/N1ICD and K5Cre$^{ERT2}$/N1ICD mice, at different timepoints after Notch1 activation (1, 3, 4, and 6-week chase) (Appendix Table S1). Conditional expression of N1ICD was triggered in SMA$^{pos}$ or K5$^{pos}$ BCs, to assess if these two basal-specific Cre drivers would target different BCs characterized by distinct degrees of differentiation or specialization (Prater et al, 2014). Comparison of BCs "reprogramming" with the two Cre lines indicated that any BC can lineage convert to HR$^{neg}$ LC upon Notch activation, regardless of their differentiation status or plasticity (Fig. EV1D).

After data pre-processing, including quality control to remove cells of low quality, a total of 474 cells were subject to further analyses. Unsupervised clustering identified five distinct cell clusters that were composed of both mutant nGFP$^{pos}$ and WT (nGFP$^{neg}$) cells. One cluster was enriched for BCs, as confirmed by their high expression of Krt5 and Krt14, and we called it the BAS cluster (Fig. EV1E). Two clusters were enriched for luminal markers: one representing luminal HR$^{neg}$ cells, expressing Krt8 and Krt19, which we termed HR$^{neg}$ cluster, and the second one, mainly composed of WT luminal mature cells, expressing Esr1 and Pgr coding for the Estrogen Receptor-α and Progesterone Receptors, named HR$^{pos}$ (for Hormone Receptor positive) cluster. Interestingly, we identified two distinct clusters, called INT1 and INT2, representing the intermediate cells that only appear upon Notch activation (Fig. 2A–C).

We then calculated basal and luminal scores based on published transcriptomic profiles of adult mammary epithelial cells (MECs) (Kendrick et al, 2008) (Figs. 2C and EV1F). As expected, both INT1 and INT2 clusters presented mixed basal and luminal scores, indicating a hybrid signature characterized by the co-expression of basal and luminal markers.

Analysis of differentially expressed genes (DEGs) (Fig. 2D) revealed that the basal markers Acta2, Sparc, and Krt14 were

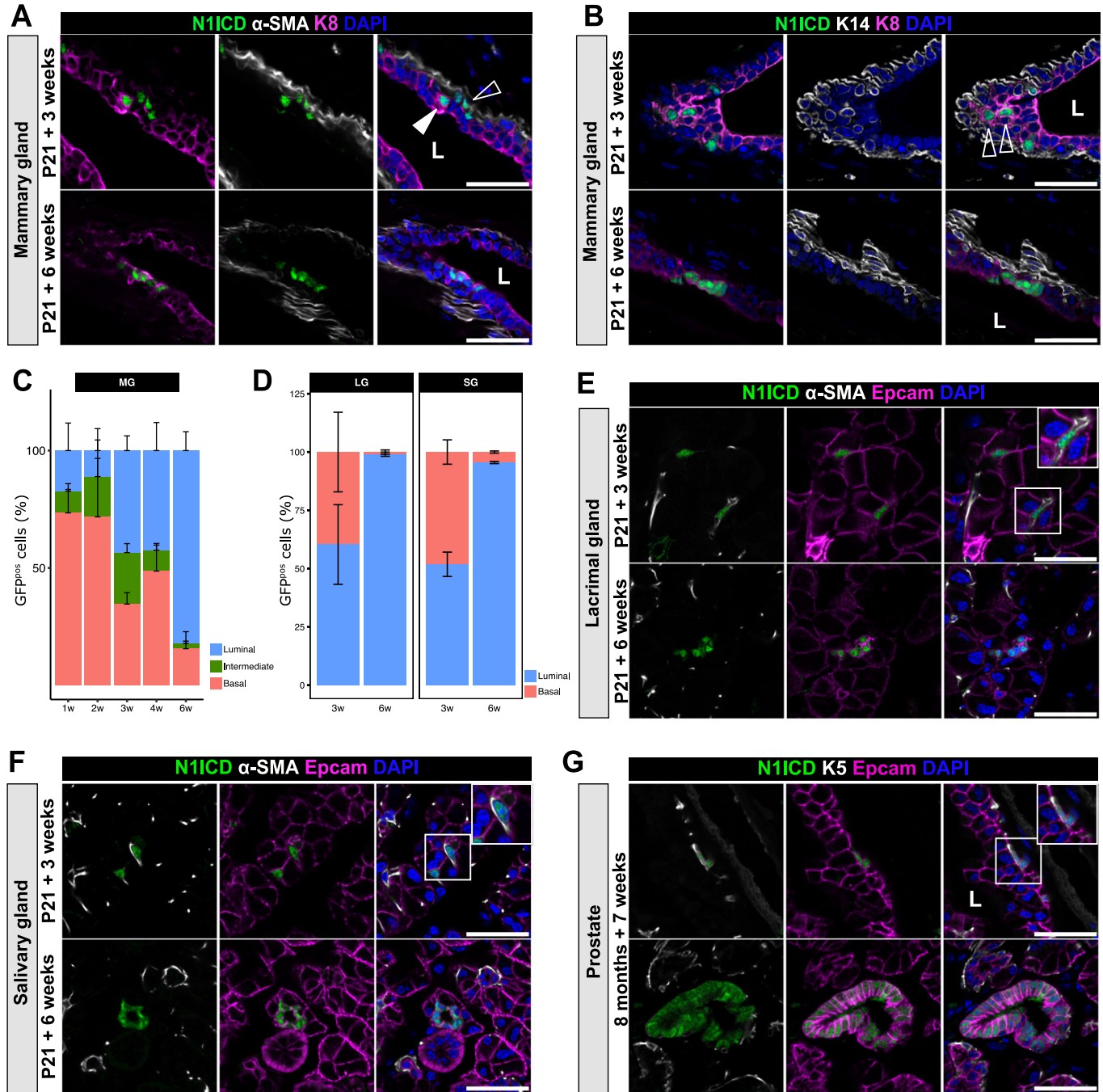

expected at high levels in the BAS cluster and were progressively reduced in INT1, whereas genes typically associated with luminal identity, such as *Plet1*, *Kit*, and *Aldh1a3*, were enriched in the HR^neg cluster and reduced in INT2. However, we could not identify genes exclusive of the INT1 cluster, and only seven genes were specific to INT2. Moreover, these genes appeared to be upregulated in a minority of cells and did not define most of the cells in this cluster. Among these genes, we found *Tacc3*, involved in the stabilization of the mitotic spindle (Ding et al, 2017; Singh et al, 2014) and *Racgap1*, required for cytokinesis (Lekomtsev et al, 2012). Importantly, we also identified a group of cells, predominantly

within the INT2 cluster, with a highly enriched cell cycle score (Fig. 2E; Appendix Table S2). When we examined the cycle profile of cells comprised in the proliferative group (Fig. EV1G) these cells were exclusively classified within S or G2/M phase. We then computed an S and G2/M molecular score for each sequenced cell and found specific enrichment of these scores in the highly proliferative cells boxed in Fig. EV1H, comprising mainly the INT1 and INT2 clusters (Fig. EV1G,H). This proliferative fraction expressed classic genes associated with cell cycle progression, such as *Mki67*, *Top2a*, and *Aurkb* (Fig. EV1I). It is also noteworthy that upon quantification of the proportion of cells from each cluster in

**Figure 1.  In vivo reprogramming of adult BCs to LCs by Notch1 activation in four bi-layered glandular epithelia.**

(A) Representative images of SMACreᴱᴿᵀ²/N1ICD mammary gland sections induced at P21 and analyzed 3 or 6 weeks later by immunofluorescence for nGFP (correlated to N1ICD expression in green), the basal marker α-SMA (white) and the luminal marker K8 (purple). Nuclei are stained with DAPI (blue). The empty arrowhead indicates basal mutant cells, the white arrowhead indicates a luminal mutant cell at 3-week chase. (B) Representative sections of SMACreᴱᴿᵀ²/N1ICD mammary glands induced at P21 and analyzed 3 or 6 weeks later by immunofluorescence for nGFP (correlated to N1ICD expression in green), the basal marker K14 (white) and the luminal marker K8 (purple). Nuclei are stained with DAPI (blue). Empty arrowheads indicate mutant cells co-expressing nGFP, K14 and K8. (C) Quantification by flow cytometry of the percentage of nGFP$^{pos}$ basal (CD49f$^{high}$/EpCAM$^{low}$), intermediate (CD49f$^{med}$/EpCAM$^{med}$) and luminal (CD49f$^{low}$/EpCAM$^{high}$) cells 1, 2, 3, 4, and 6 weeks after tamoxifen induction at P21 (bars represent mean ± SEM) in MG. $n = 6$. (D) Quantification by immunofluorescence on sections of the percentage of GFP$^{pos}$ cells classified as basal (αSMA$^{pos}$) or luminal (EpCAM$^{high}$) 3 or 6 weeks after activation of N1ICD in LG and SG. Bars represent mean ± SEM. $n = 3$. (E, F) Representative sections of SMACreᴱᴿᵀ²/N1ICD lacrimal glands (E) or salivary glands (F) induced at P21 and analyzed 3 or 6 weeks later by immunofluorescence for nGFP (N1ICD in green), the basal marker α-SMA (white) and the luminal marker EpCAM (purple). Nuclei are stained with DAPI (blue). Insets illustrate magnifications of the white boxed areas. (G) Representative sections of K5Creᴱᴿᵀ²/N1ICD prostate induced at 8 months and analyzed 7 weeks later by immunofluorescence for nGFP (N1ICD in green), the basal marker K5 (white), and the luminal marker EpCAM (purple). Nuclei are stained with DAPI (blue). Inset illustrates magnifications of the white boxed areas. Scale bar represents 25 μm in (A, B, E–G). "L" indicates the lumen position in (A, B, G).

G1, S, or G2/M, we found that the highly proliferative INT2 cluster comprises fewer cells in G1 and more in the G2/M phase compared to all other clusters (Fig. EV1J).

A more refined analysis of the luminal cluster containing both WT (nGFP$^{neg}$) and mutant (nGFP$^{pos}$) LCs revealed their segregation based on their mutant status, suggesting that, even if they are identified as LCs by FACS, nGFP$^{pos}$ mutant cells remained somehow different from fully differentiated LCs (Fig. 2A). Consistent with this, Principal Component Analysis (PCA) confirmed that mutant luminal cells (GFP$^{pos}$) at 3 and 4 weeks of chase do not entirely overlap with WT luminal (GFP$^{neg}$) (Figs. 2A and EV2A). To reveal the differences between GFP$^{neg}$ and GFP$^{pos}$ luminal cells, we performed UMAP analysis exclusively within the HR$^{neg}$ luminal cell cluster, and we could distinguish two subclusters. Most GFP$^{neg}$ cells were associated with one cluster, that we named luminal cells, whereas the other cluster was mainly composed of GFP$^{pos}$ mutant cells, that we called pre-luminal cells. The quantification of the percentage of GFP$^{pos}$ and GFP$^{neg}$ cells in the two clusters indicates a predominance of mutant cells in the pre-luminal group (Fig. EV2B). These two clusters are mainly distinguished by the time after N1ICD activation, as cells seem to acquire a pre-luminal identity at 1, 3, and 4 weeks and a more complete luminal identity after 6 weeks (Fig. EV2C). Although well-established luminal markers, such as *Krt18* and *Epcam* (Fig. EV2D), were similarly expressed by these two clusters, other luminal genes, like *Trf*, *Muc1*, and *Clic6*, presented very low levels of expression in the pre-luminal cluster, corroborating the notion that these cells, identified as luminal cells based only on their elevated expression of the EpCAM epithelial marker, have not entirely acquired a luminal identity (Fig. EV2E). To further highlight differences between WT and mutant HR$^{neg}$ luminal cells, we selected the cells index-sorted as luminal HR$^{neg}$ by FACS and performed DEG analysis. This query identified some genes differentially expressed in WT and mutant cells (Fig. EV2F). Notably, the gene *Dclk1*, encoding for Doublecortin-like kinase 1, and associated with gastric and breast cancer (Afshar-Sterle et al, 2024; Wang et al, 2019), was found uniquely and highly upregulated in mutant luminal HR$^{neg}$ cells. On the contrary, *Igfbp5*, the gene encoding the Insulin-like growth factor-binding protein 5, was strongly enriched in WT HR$^{neg}$ cells. This gene has been reported to be causally related to apoptosis in the mammary gland (Allan et al, 2004), although the significance of its high expression in this context remains to be established.

It is important to consider that mutant LCs cannot be considered WT LCs even once the lineage switch is complete, since they continue to express N1ICD. This perduring Notch

activity results in the formation of mammary hyperplastic lesions (Fig. EV2G,H) after a long latency (8 months) or upon 1 or 2 cycles of pregnancy and lactation, that can occasionally become malignant tumors, as previously observed (Bouras et al, 2008; Ling and Jolicoeur, 2013). Based on our computed basal and luminal scores and on the list of DEGs, the two clusters of intermediate cells expressed a mixed gene set between luminal and basal markers, with INT1 more closely related to the BAS cluster and INT2 more luminal, suggesting a progressive transcriptional switch from a basal to a luminal differentiation program.

Given that several single-cell transcriptomic studies have described a population of cells co-expressing basal and luminal markers (often called hybrid cells) in embryonic mammary glands (Giraddi et al, 2018; Pal et al, 2021; Wuidart et al, 2018) or in response to adult LCs ablation (Centonze et al, 2020), we wondered if the INT clusters we identified in our study reflected the presence of cells that reactivated embryonic or regenerative programs typical of multipotent MaSCs. To interrogate this, we compared our intermediate cells with hybrid cells identified in published datasets using Label Transfer from the Seurat package (Stuart et al, 2019), a variant of integration which allows the transfer of cluster labels from a reference dataset to a query dataset. The scRNAseq profile of mammary cells published by Wuidart and colleagues (Wuidart et al, 2018) comprised adult BCs and LCs, as well as "Embryonic Multipotent Progenitors" (EMPs) (CD49f$^{high}$/Lgr5-GFP$^{high}$) isolated from the mammary epithelium at embryonic day 14 (E14). EMPs co-expressed genes typical of BCs and LCs, but they also presented specific genes that were defined as the EMPs signature. By integrating our dataset with the Wuidart et al, dataset, we found, as expected, that the basal cell cluster (BAS), as well as the HR$^{neg}$ and HR$^{pos}$ clusters, overlapped with the same adult cell types identified in that study (Fig. EV3A). Of interest, the INT1 and EMPs clusters broadly co-localized in the PCA plots, validating their hybrid signature, whereas INT2 appeared as a separate cluster that was not identified by Wuidart and colleagues. Using label transfer with the dataset by Wuidart *et al*, as a reference and visualizing the transferred labels with an alluvium plot, we found that while the INT1 cluster is splitted between BAS and HR$^{neg}$ adult clusters, INT2 is almost entirely linked to the HR$^{neg}$ WT cluster and clearly diverges from the BAS cluster (Fig. EV3B). To our surprise, however, this comparative analysis indicated that the EMPs signature was not specifically expressed in INT1 or INT2 clusters (Fig. EV3C). This comparative analysis indicated that the intermediate cells that we

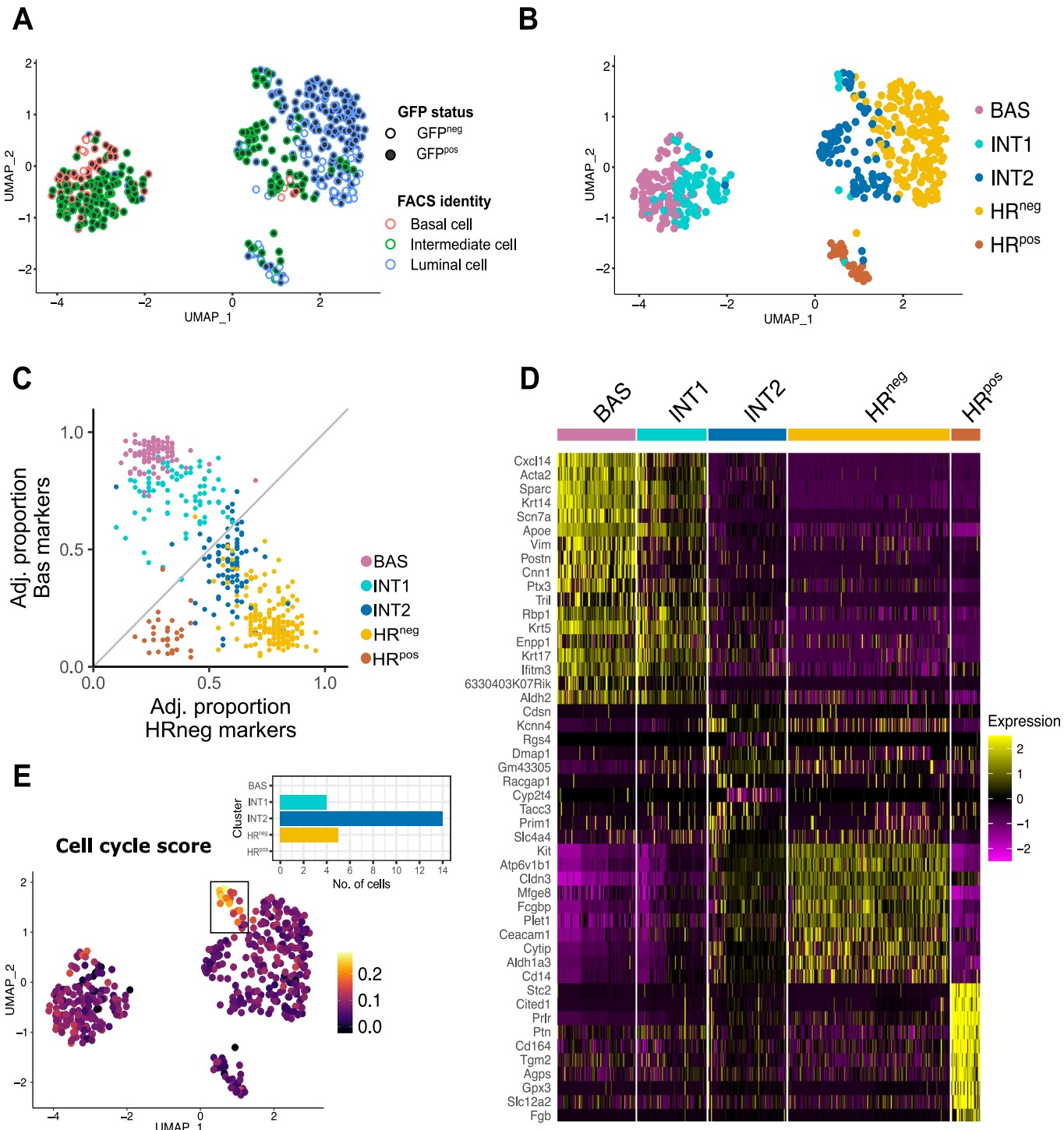

**Figure 2. Index-sorted single-cell RNAseq reveals the hybrid signatures of transitioning intermediate mutant cells.**

(A) UMAP plot showing the identity of each index-sorted cell along with their GFP status. Cells are color-coded based on their FACS-defined identity as Basal (red), Intermediate (green), and Luminal (blue) cells. Mutant cells (GFP$^{pos}$) are depicted as filled dots; WT cells (GFP$^{neg}$) are shown as empty dots. (B) UMAP plot showing clustering of single sequenced cells by Smart-seq2. 5 Seurat clusters were identified: BAS basal cells (pink), INT1 intermediate 1 (turquoise), INT2 intermediate 2 (dark blue), HR$^{neg}$ hormone receptor$^{neg}$ (yellow), and HR$^{pos}$ hormone receptor$^{pos}$ cells (brown). BAS = 89 cells, INT1 = 79 cells, INT2 = 89 cells, HR$^{neg}$ = 184 cells, HR$^{pos}$ = 33 cells. (C) Scatter plot representing the adjusted proportion of basal markers found in the BAS cluster (y-axis) and of luminal markers detected in the HR$^{neg}$ cluster (x-axis). Every dot represents a cell, and the colors reflect the clusters defined in Fig. 2B. (D) Heatmap showing the expression levels of genes specific for each cell cluster illustrated in (B). Each column is color-coded according to the corresponding cell cluster from (B). The color key corresponds to normalized and scaled values of gene expression. (E) UMAP plot showing enrichment for the GO term "Cell cycle" across individual cells. The bar plot represents the number of cells, grouped by cluster, within the rectangular selected region in the UMAP representation.

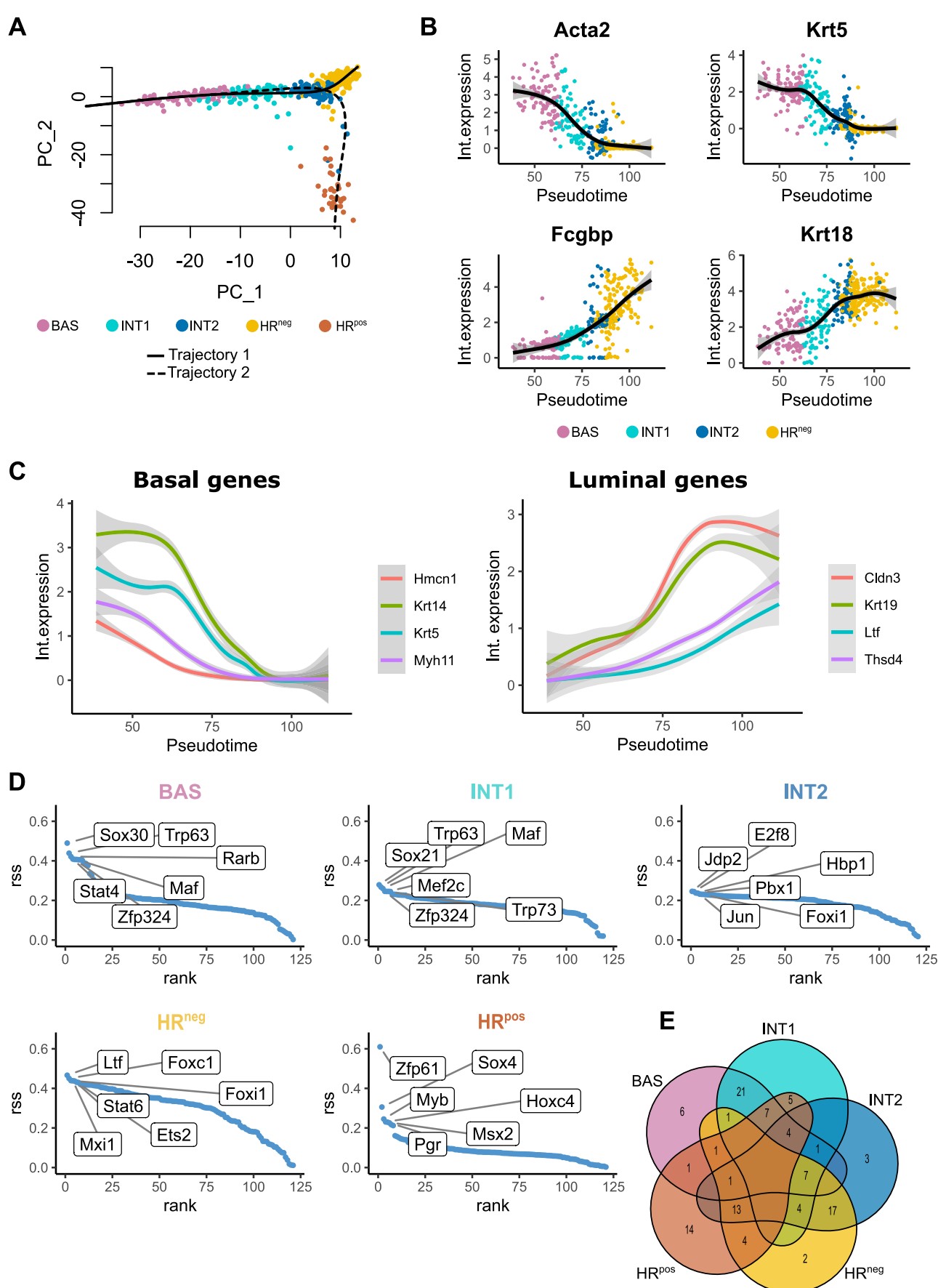

**Figure 3. Lineage trajectory and transcriptional signatures defining the progressive transition from basal to luminal identity.**

(A) Slingshot trajectory analysis showing two cellular paths, connecting BAS cells to HR$^{neg}$ (trajectory 1) or HR$^{pos}$ (trajectory 2) clusters in a PCA plot. (B) Expression of selected genes within cells plotted along trajectory 1 in pseudotime. The integrated gene expression is plotted; dots correspond to individual cells color-coded according to the UMAP clusters from Fig. 2B. (C) Expression of selected basal and luminal genes along pseudotime trajectory 1. (D) SCENIC analysis showing the Regulon specificity score (RSS) for each cluster: only the six most significant TF regulons showing cluster-specific activity are indicated. (E) Venn diagram presenting the number of overlapping regulons among the 50 most significant TF regulons for each cell cluster.

identified in our dataset do not necessarily revert to an embryonic multipotent progenitor state similar to the EMPs sequenced at embryonic day E14 by Wuidart and colleagues.

We then integrated our dataset with the sequencing results from Centonze et al, (Centonze et al, 2020), who performed scRNAseq of adult BCs and LCs following genetic ablation of a fraction of LCs in vivo. This genetic intervention induced reactivation of multi-potency in BCs and the appearance of a population of hybrid cells (referred to as "Hybrid") showing co-expression of basal and luminal markers. PCA of the integrated datasets indicated that our INT1 cluster closely associates with the Hybrid cells and partially overlaps with the adult BAS cluster, whereas, once again, the INT2 cluster stands alone, featuring little overlap with the Hybrid cells sequenced by Centonze and colleagues (Fig. EV3D). The analysis of the same dataset using the alluvium plot representation confirmed that our INT1 cluster is transcriptionally closer to BAS and Hybrid cells from Centonze et al, whereas INT2 cells appear more closely related to HR$^{neg}$ cells (Fig. EV3E).

This comparative in silico analysis suggests that the INT1 cluster represents a hybrid transcriptional cell state in between basal and luminal lineages, similarly to the Hybrid cluster reported by Centonze et al. On the contrary, the INT2 cluster is uniquely found upon Notch activation and, as such, it may represent a distinctive cluster, possibly more related to committed luminal cells.

To further characterize the intermediate transcriptional states that we have identified and assess if they can be found in physiological mammary glands at specific times of major tissue remodeling, such as in puberty or pregnancy, we compared our different clusters with datasets obtained from terminal end buds (TEBs) or ductal cells at puberty (Pal et al, 2021). Label transfer analysis showed that basal cells in TEBs (usually referred to as cap cells) were more similar to the INT1 cluster than to our BAS cluster (Fig. EV3F). Likewise, the HR$^{neg}$ TEB cells corresponded to the INT2 cluster instead of pairing with our HR$^{neg}$ cluster. These results suggest that the intermediate cell states we have identified in this study resemble the less differentiated pubertal TEB cells which have not yet acquired the full repertoire of luminal or basal definitive markers of mature cells. This finding is corroborated by the fact that pubertal ductal BCs indeed cluster with our BAS cells. However, the HR$^{neg}$ ductal cells in puberty still correlated to our INT2 cluster rather than to the more differentiated HR$^{neg}$ cells. This might indicate that pubertal ductal BCs are more mature than the cap cells in TEBs, thus they correspond to adult BCs, whereas HR$^{neg}$ pubertal cells, even in ducts, are less mature than adult HR$^{neg}$ LCs.

In addition, we compared our dataset with sequencing results obtained in pregnant mice (Bach et al, 2017), and found that basal Bsl-G (Basal cells at gestation) were similar to our BAS cells, whereas Avd (alveolar differentiated cells) corresponded in part to our HR$^{neg}$ and some to our INT2 (Fig. EV3G). These findings lead us to conclude that the Avd cluster comprises both differentiating

alveolar cells (the fraction pairing with our INT2 cells) and more differentiated alveolar cells that correspond to our HR$^{neg}$ luminal cluster.

In conclusion, the integration of our single cell profiles with published datasets indicates that the intermediate cells that appear upon Notch ectopic activation, although featuring a mixed signature, are different from embryonic multipotent MaSCs, but they represent less differentiated cell states than adult BCs or LCs, resembling pubertal cells found in TEBs.

## Transcriptional landscapes underlying the progressive lineage transition from BCs to LCs

To examine the gradual transcriptional changes that occur during the cell fate switch from BCs to LCs, we then performed a slingshot trajectory analysis within the PCA space, denoting the BAS cluster as the origin. Interestingly, we observed a forked pattern presenting 2 separate trajectories (Fig. 3A). Both trajectories passed through the INT1 and INT2 clusters, but one path terminates in the HR$^{neg}$ cluster (Trajectory 1) while the other ends with the HR$^{pos}$ cluster (Trajectory 2). The divergence of the two trajectories was observed around the INT2 cluster, suggesting that the two luminal identities are specified at the latest intermediate stage.

Given that Notch1 activity is restricted to ERα$^{neg}$/PR$^{neg}$ luminal cells and that the cell fate switch induced by Notch activation eventually converts the targeted BCs exclusively into ERα$^{neg}$/PR$^{neg}$ LCs, we then focused our analysis on trajectory 1 (BAS to HR$^{neg}$). For this, we used Tradeseq, which performs Generalized Additive Model (GAM) fitting to the gene expression variation along pseudotime. During the transition from BAS to HR$^{neg}$ fate, we confirmed the previously detected trend of progressive decrease in expression of classical basal markers, such as Acta2 and Krt5, and gradual increase in luminal gene expression, including Krt18 and Fcgbp (Figs. 3B and EV4A). In general, most genes followed a trend of constant increase or decrease of expression during the progressive switch from basal to luminal cell identity. Along the pseudotime, we noticed that the first event detectable at the transcriptional level consists of the downregulation of basal genes, as we had previously found in bulk RNAseq experiments (Lilja et al, 2018), and this is associated with cells belonging to the INT1 cluster. Later on along the pseudotime, the expression of luminal markers kicks in, in cells belonging to the INT2 cluster, which present co-expression of several genes typical of either BCs or LCs, such as Krt17 and Cldn3, most likely corresponding to the K14/K8 double-positive cells that we observed by immunostaining (Fig. 1A) (Lilja et al, 2018). Finally, the last step of the transition is characterized by the complete loss of basal genes and the steady increase of expression of luminal lineage genes (Fig. EV4A,B).

The pseudotime analysis we performed clearly indicates the progressive nature of the lineage transition induced by ectopic Notch1

activation, characterized by continuous and gradual transcriptional changes underlying the sequential change in cell identity. We thus conclude that lineage conversion requires a stepwise and asynchronous change in transcriptional programs, with some basal genes downregulated early and others that take a longer time to be repressed. For example, the gene *Hmcn1*, encoding the immunoglobin superfamily member Hemicentin 1 and the smooth muscle myosin heavy chain *Myh11*, a well-described marker of the basal lineage (Prater et al, 2014), are among the earliest basal genes to be downregulated, whereas the typical basal cytokeratins *Krt14* and *Krt5* decrease in expression only later (Fig. 3C). The same is true for the acquisition of a luminal identity, with genes such as *Krt19* and *Cldn3* that are upregulated very early during the transition, and others, like *Ltf* and *Thsd4*, whose increase in expression is only observed toward the end of the transition in pseudotime (Fig. 3C).

We next sought to identify differentially expressed genes for each mammary epithelial cluster by examining their dynamic expression profile towards the lineage switch trajectories. This extended analysis identified 6 different patterns of expression along the process of basal to luminal transition (Fig. EV4B; Appendix Table S3). We identified two groups enriched in basal genes ("bas early" and "bas late"), which include well-known basal cell markers such as *Krt15* and *Krt5*, *Sparc* or *Cxcl14* and are linked to the GO term "extracellular matrix organization" (Fig. EV4C). Likewise, toward the end of the pseudotime, we found enrichment in the expression of genes associated with the acquisition of the luminal phenotype ("lum early" and "lum late"), including *Krt8*, *Clic6*, or *Aldh1a3*.

Of interest, this analysis highlighted two patterns of expression specifically associated with the intermediate clusters (that we called "int early" and "int late"). The "int early" group presented an enriched expression of the cell adhesion genes *Sdc1* and *Nebl*, whereas genes associated to cell proliferation, including *Mcm2* and *Ccnd1*, defined the "int late" group (Appendix Table S3).

The dynamic expression profile of genes associated with these six patterns of expression indicates that the transition from basal to luminal identity is a long and progressive process involving the following sequence of events: initial repression of typical basal genes, modification of cell adhesion properties, proliferation, progressive acquisition of a luminal phenotype, and finally complete expression of definitive luminal markers.

## Gene regulatory network analysis uncovers the molecular signatures of transitioning cells

To further capture the regulatory mechanisms at work during the cell fate switch, and to identify potential transcriptional nodes that could represent general regulators of cell plasticity, we then performed SCENIC analysis on our dataset. The SCENIC algorithm examines the activity of transcription factor regulons, consisting of transcription factors and their targets, within individual cells (Aibar et al, 2017). We used the regulon specificity score (RSS) to identify regulons showing enriched activity in each cell cluster. We observed, as predicted, elevated activity of the progesterone receptor (*Pgr*) regulon in HR[pos] luminal cells and of *Foxc1* in HR[neg] luminal progenitors, consistent with a previous report (Sizemore et al, 2013) (Fig. 3D). Among the top 50 regulons enriched in cluster INT1, we could not pinpoint any that was exclusive for this cluster and was not shared with other clusters, and most of these regulons were found in both INT1 and BAS clusters, indicating a strong overlap of

INT1 and basal cell identity (Fig. 3E). Likewise, most of the regulons enriched in cluster INT2 were shared with the HR[neg] cluster. This analysis corroborates our findings, indicating that the early steps of lineage switch involve suppression of basal regulons, such as *Trp63* and *Trp73*, followed by the steady and progressive increase of activity of luminal-specific regulons, such as *Jun* or *Stat6* (Fig. EV4D). Consistent with our results, when we analyzed for cluster-specific regulon activity, we could not identify regulons that would be unique to INT1 cells. We could, however, find three regulons, *Brca1*, *E2f8*, and *E2f1*, which were specific to INT2 cells, and these are all linked to elevated proliferation. Interestingly, these regulons are enriched in highly proliferative cells belonging to the INT1, INT2, and HR[neg] clusters (Fig. EV4E).

This analysis suggests that the lineage conversion involves activation of a proliferative signature, particularly in cells belonging to the INT2 cluster, for mutant cells to complete the fate transition and engage into transcriptional programs characteristic of luminal cells.

## Proliferation is indispensable for switching cell identity

The scRNAseq analysis on index-sorted mutant cells undergoing the cell fate transition allowed us to identify a group of cells, mainly belonging to the INT2 cluster, characterized by high cell cycle score and upregulation of regulons linked with active proliferation. In addition, when we triggered Notch activation in pre-pubertal (3-week-old) or adult (10-week-old) mice, we found that adult mammary cells take longer to complete the transition from basal to luminal identity. While induction before puberty (at postnatal day P21) resulted in a minimum of 85% of mutant LCs within 6 weeks, in adult mice, we could detect about 30% of BCs that had not yet switched fate after a 6-week chase (Fig. EV5A). To understand these differences in the kinetics of lineage conversion, we quantified the proportion of Ki67-positive cells in pubertal and adult mice and found that pubertal mammary epithelial cells (MECs) proliferate more frequently than adult cells (Fig. EV5B), thus they convert to luminal fate more rapidly than the more quiescent adult mammary cells. Of note, this difference could not be explained by recombination rate, since we found no variance in the proportion of GFP-labeled cells at puberty or in adult mice upon a 48 h tamoxifen pulse (Fig. EV5C). Based on the scRNA-sequencing analysis and these in vivo results, we formulated the hypothesis that the switch in cell identity does not simply represent a transdifferentiation event, bypassing cell division, but rather requires actively proliferating cells that respond to Notch activation by giving rise to luminal daughter cells.

In order to experimentally test if proliferation was required for the transition from BCs to LCs induced by ectopic Notch1 activation, we thus implemented the culture of 3D organoids from epithelial cells of the mammary gland, the SG, LG, and prostate (Charifou et al, 2021; Jardé et al, 2016). First, we established that this in vitro system was suitable to study the cell fate switch, by demonstrating that WT cells derived from the adult mammary epithelium maintain a unipotent behavior in the organoid culture conditions. Indeed, lineage tracing of WT BCs in organoids, using SMACre[ERT2]/mTmG (for mammary gland, SG and LG) and K5Cre[ERT2]/mTmG (for the prostate) mice revealed that exclusively basal daughter cells were derived from the initially labeled BCs and were therefore marked by our lineage tracer

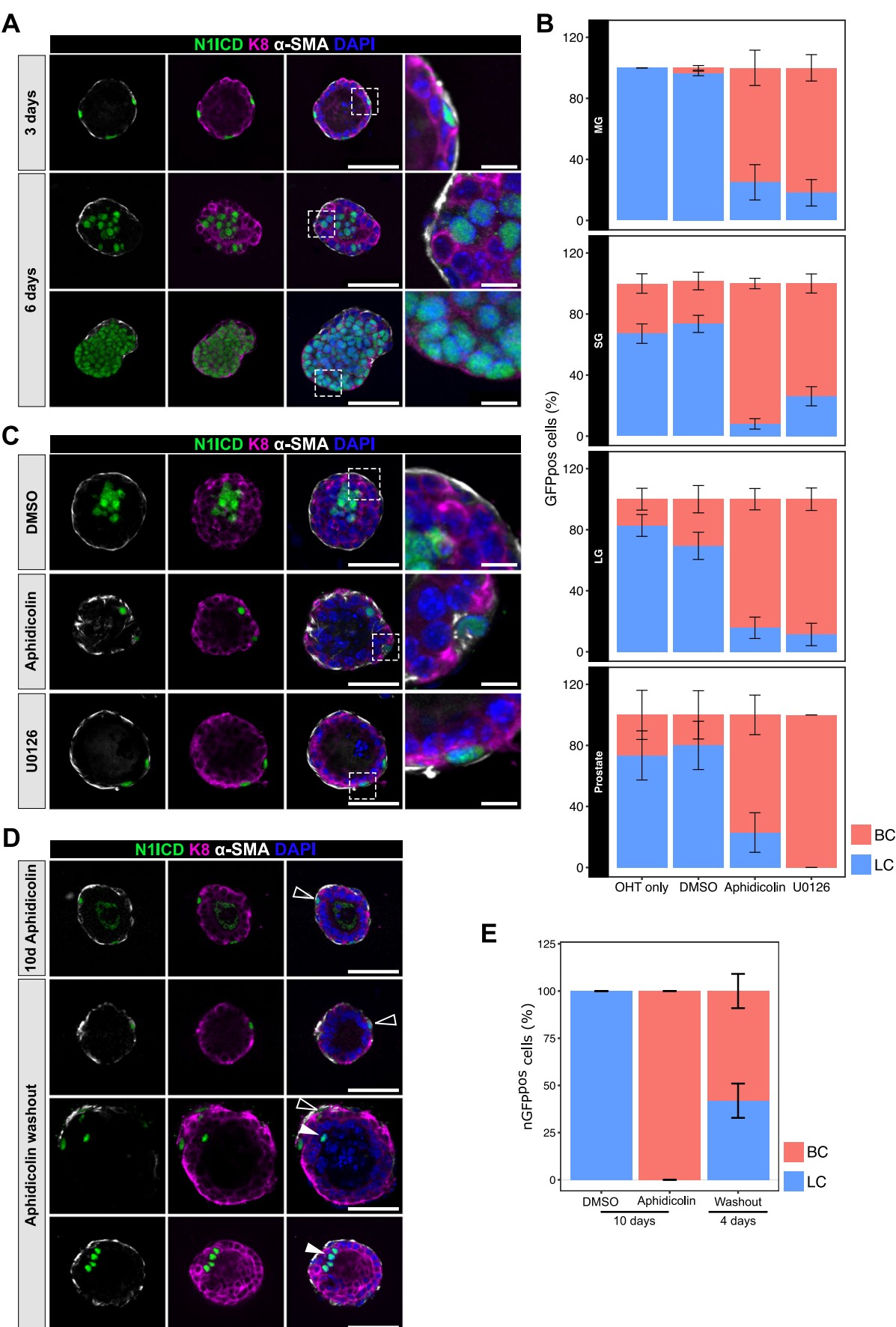

**Figure 4. Proliferation is an obligatory step for lineage transition in organoids.**

(A) Representative images showing immunofluorescence for nGFP (N1ICD in green), luminal K8 (purple), and basal α-SMA (white) expression in SMACre^ERT2^/N1ICD mutant mammary organoids 3 or 6 days after 4-OHT induction. (B) Quantification of the proportion of nGFP⁺ BC (basal in red) or LC (luminal in blue) mutant cells in mammary gland (MG), LG, SG, and prostate, in OHT only, OHT + DMSO, OHT + Aphidicolin, or OHT + U0126 for 6 days. Data were displayed as mean ± SEM. This graph represents three independent experiments and a minimum of six organoids. (C) Representative images showing immunofluorescence for nGFP (N1ICD in green), luminal K8 (purple), and basal α-SMA (white) expression in SMACre^ERT2^/N1ICD mutant mammary organoids treated with DMSO, Aphidicolin or U0126 for 6 days and after 4-OHT induction. At least five organoids were counted per condition. (D) Representative images showing immunofluorescence for nGFP (N1ICD in green), luminal K8 (purple), and basal α-SMA (white) expression in SMACre^ERT2^/N1ICD mutant mammary organoids treated with Aphidicolin for 6 days and grown for another 4 days upon Aphidicolin washout or treated with Aphidicolin for 10 consecutive days. Nuclei are stained with DAPI in blue. The scale bar represents 50 μm in (A, C, D) and 10 μm for the magnified insets (in A, C). Empty arrowheads indicate cells that have not undergone cell fate switch at the time of the analysis, white arrowheads indicate nGFP^pos^ luminal cells (in D). (E) Quantification of the proportion of nGFP^pos^ BC (in red) or LC (in blue) mutant cells within each organoid after 10 days of DMSO or aphidicolin treatment or after Aphidicolin washout for 4 days. Data were displayed as mean ± SEM. This graph represents three independent experiments and a minimum of five organoids.

membrane-bound mGFP (Fig. EV5D). Importantly, upon in vitro Notch1 activation via 4-hydroxitamoxifen (4-OHT) administration to the organoid medium, we could recapitulate the progressive transition of mutant nGFP^pos^ cells from the basal to the luminal lineage, also correlated with the progressive increase in expression of K8 and loss of α-SMA, robustly reflecting the data acquired in vivo (Figs. 4A and EV5E). This transition was also recapitulated in prostate, lacrimal, and salivary gland organoids with very similar kinetics (Figs. 4B and EV5F). Remarkably, a complete cell fate switch in organoids could be achieved within only 6 days after induction of N1ICD expression. Thus, the organoid system allows us to induce a rapid cell fate switch, much faster than in vivo, and to target more cells, such that some organoids were exclusively composed of luminal nGFP^pos^ cells 6 days after Cre induction (Fig. 4A,B).

Our in vivo data, corroborated by the single cell transcriptional analysis at different timepoints after Notch activation, demonstrated that the lineage transition is asynchronous, with some cells switching to a luminal fate more rapidly than others, indicating a heterogeneous competence of different BCs to respond to Notch activation. Given that resting adult mammary cells take longer to switch than proliferating pubertal cells and that organoid cells switch even more rapidly, we then investigated the involvement of proliferation in dictating the differential readiness of BCs to transition towards a luminal fate. Validating our hypothesis, pharmacological block of proliferation in organoids, confirmed by EdU staining (Fig. EV5G,H), by supplementing the culture medium with either Aphidicolin, an inhibitor of DNA polymerase, or U0126, a MAPK inhibitor, led to a significant impairment of the cell fate switch (Fig. 4B,C), contrary to DMSO-treated control organoids, where the lineage transition was generally completed within 6 days. When we quantified EdU incorporation 3 days after OHT administration, we found that a higher proportion of mutant BCs was proliferative, compared to WT BCs (Fig. EV5I), indicating that Notch activation promotes proliferation, obligatory for the lineage conversion. Remarkably, proliferation was also essential for the cell fate switch of SG, LG, and prostate cells (Figs. 4B and EV6A–C). To confirm that the observed block of cell fate switch was indeed directly associated with proliferation arrest, we then removed Aphidicolin after 6 days of treatment. Four days after washout, we found that 53% of the nGFP^pos^ cells re-entered the fate transition program and became luminal within 6 days (Fig. 4D,E). It is noteworthy that the heterogeneous behavior of mutant nGFP^pos^ cells could be observed even within the same organoid, with some mutant cells readily switching to luminal fate upon aphidicolin

washout (white arrowheads in Fig. 4D) and others more refractory to enter the lineage transition (empty arrowheads in Fig. 4D). These experiments demonstrate that the arrest in cell fate switch can be reversed (Fig. 4E), and they corroborate our findings implicating proliferation as an obligatory step for lineage conversion.

We then assessed the temporal dynamics of the fate transition in organoids, and for this, we used SMACre^ERT2^/mTmG/N1ICD compound mice, allowing us to track mutant cells by following the fluorescence of membrane-tagged GFP (mGFP from the mTmG allele) in real-time. In fact, the nGFP co-expressed with the N1ICD allele is not detectable by live microscopy, as it requires immunostaining with anti-GFP antibodies. In this experimental setting, we could identify and track by time-lapse microscopy mGFP^pos^ cells, initially localized in the basal compartment, that entered cell cycle while moving to a luminal internal position (Fig. 5A). Some mGFP^pos^ cells instead remained in the basal compartment even after mitosis (Fig. 5B, arrowheads), most likely representing WT BCs that had only floxed the mTmG reporter but not the more refractory N1ICD allele. To provide evidence that the N1ICD allele is recombined at a much lower frequency than the mTmG allele, we induced recombination for 48 h and quantified the percentage of GFP^pos^ BCs in SMACre^ERT2^/N1ICD and SMACre^ERT2^/mTmG mice. These experiments indicated that the percentage of nGFP-labeled cells was around 1–2%, whereas mGFP-positive cells represented about 40% of BCs (Fig. EV6D).

Given the observed lack of complete overlap between mGFP^pos^ cells (from the mTmG allele) and nGFP^pos^ cells (N1ICD-expressing), we quantified the proportion of mGFP^pos^ cells that also expressed the Notch1 direct target Hes1 in mammary organoids. While we found that around 50% of the mGFP^pos^ cells remained in a basal position even after 6 days of induction, these cells invariably lacked Hes1 expression (Fig. 5C). Importantly, we could not detect any mGFP^pos^/Hes1^neg^ luminal cells, demonstrating that these BCs were indeed WT, hence retained unipotency and did not generate mGFP^pos^ luminal daughters (Fig. 5C,D). The time-course analysis at 3 and 6 days after induction clearly illustrates that the proportion of mGFP^pos^/Hes1^pos^ luminal cells increases, representing lineage-switching mutant cells, concomitantly with the decrease in the percentage of mGFP^pos^/Hes1^pos^ basal cells.

In conclusion, we demonstrated that the lineage switch induced by Notch1 is achieved through a progressive change in cell identity, whereby mutant cells transit through an intermediate metastable state, that requires their capacity to enter mitosis.

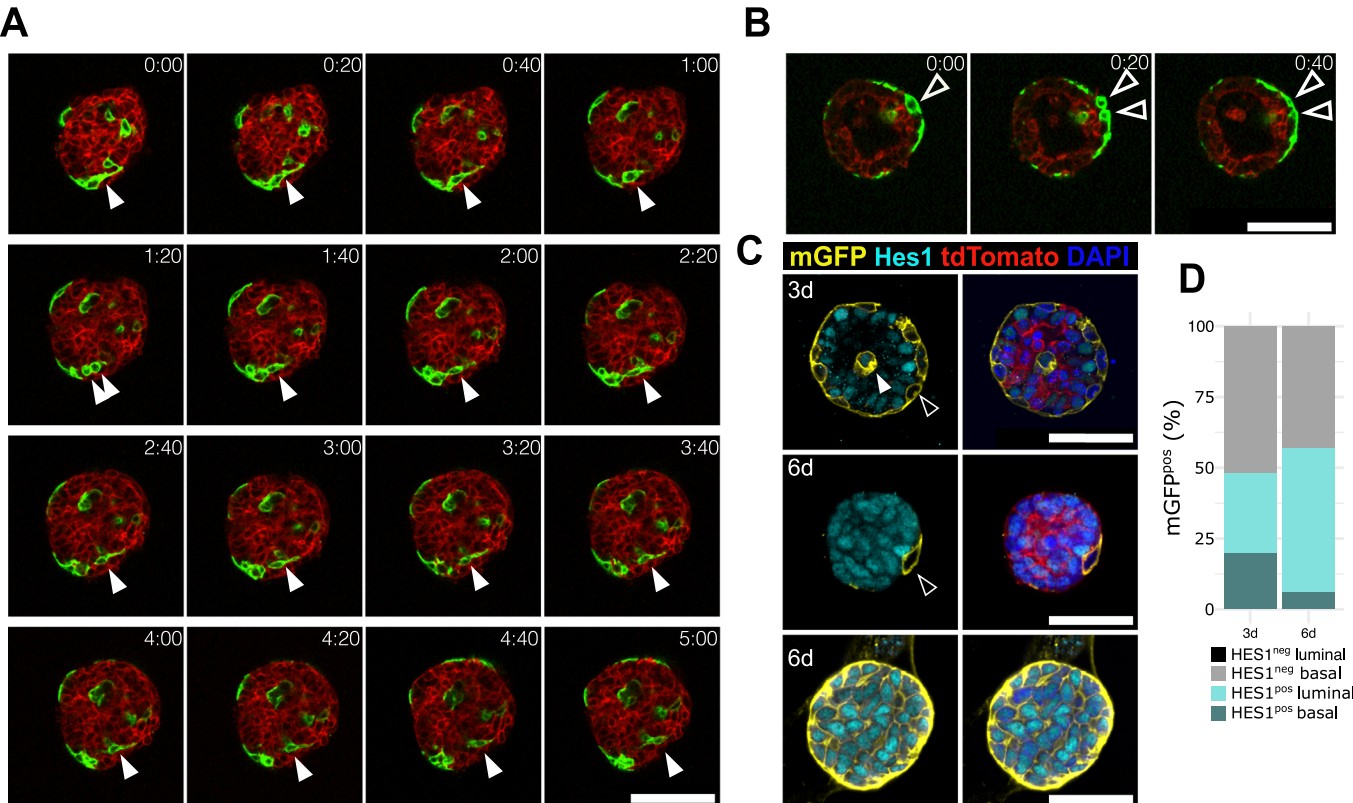

**Figure 5. Dynamic behavior of lineage transitioning cells by time-lapse analysis.**

(A, B) Sequential time-lapse images of SMACre^ERT2/mTmG/N1ICD organoids showing recombined GFP^pos (green) cell rearrangements over 5 h. Red: non-recombined tdTomato^pos cells. White arrowheads in (A) pinpoint a mutant BC that first divides (between 1 h 20 min and 1 h 40 min time frames), and then one of the two daughter cells moves to a luminal position after mitosis. The empty arrowheads in (B) depict the mitosis of a WT basal cell whose daughters stay in the basal outer cell layer after division. Scale bar represents 25 μm. (C) Representative images showing immunofluorescence for mGFP (indicating recombined cells in yellow), Hes1 (marking nuclei and reflecting Notch activation in turquoise), and tdTomato expression in red in organoids grown for 3 days. Nuclei are stained with DAPI in blue. White arrowhead indicates mutant cells (Hes1 positive) and empty arrowhead indicate WT cells (Hes1 negative). Scale bar 50 μm. (D) Quantification by immunofluorescence of the proportion of mGFP^pos basal or luminal cells classified based on their expression of HES1 and analyzed 3 or 6 days after 4-OHT induction. For the organoids composed exclusively of Hes1^pos cells, we classified all cells as luminal, based on the analysis performed in Fig. 4A.

## Discussion

We report here that Notch signaling is a gatekeeper of luminal cell fate and that its critical role of dictating binary cell fate choices is conserved in several tissues, as demonstrated by the fact that ectopic Notch1 activation in committed adult BCs reprograms them toward a luminal identity in four different glandular epithelia. Importantly, we observed both in vivo and in organoids that BCs ectopically induced to activate Notch signaling rapidly move towards the ductal lumen, while acquiring luminal characteristics, indicating that intrinsic signals dictate cell fate, leading to positional changes and cell rearrangements within bi-layered branched epithelia. Future studies will be required to probe if Notch activity directly influences cell position and movements or if other factors act on cell dynamics and contribute to establishing the definitive commitment toward a luminal cell fate.

We demonstrate here that the transition from basal to luminal state is achieved through a progressive transcriptional switch triggered by the initial downregulation of basal genes, followed by the upregulation of luminal differentiation programs. These two cellular states are not exclusive, as demonstrated by the presence of

hybrid cells co-expressing luminal and basal markers (K14^pos/K8^pos cells). While the presence of hybrid cells has been reported in several contexts, and is believed to reflect the remarkable cellular plasticity of mammary BCs, it was not known that proliferation is a mandatory step to induce this intermediate metastable cell state and to accomplish the lineage switch to LCs. These findings indicate that adult mammary BCs, when forced to activate Notch signaling and change fate, do not undergo transdifferentiation, but rather that they are reprogrammed to a plastic state that, despite their initial unipotency, enables them to give rise to LCs, thus alters their differentiation potential independently of their position within the tissue. Mutant BCs appear to transition through a transient intermediate phase of co-expression of basal and luminal markers before attaining a luminal identity and eventually giving rise exclusively to fully 'reprogrammed' LCs. We show that BCs initially reduce the expression of basal genes, and then they enter a state of active proliferation, which results in the generation of luminal daughter cells. This behavior reflects their extensive plasticity and does not necessarily require Notch activity, since WT BCs induced to reactivate bipotency by LCs genetic ablation also require proliferation to give rise to new luminal daughters, as

shown by the fact that decreasing proliferation using a CDK1 inhibitor or by p21 overexpression in mammary organoids impaired BCs multipotency (Centonze et al, 2020). The hybrid cell state, characterized by the co-expression of basal and luminal genes within the same cell, is also often found in breast cancer, and it has been associated to a multipotent state (Van Keymeulen et al, 2015; Koren et al, 2015). However, the continuous expression of active Notch1 in our model prevents the maintenance of multipotent stem cells since it forces cell differentiation toward a luminal fate, and eventually, all mutant cells become HR[neg] luminal cells.

Differentiation and proliferation are usually two cellular anticorrelated processes. However, the ectopic activation of Notch1 in differentiated BCs could induce the expression of cytokine-related cell cycle genes, like *Cdk1* (Ronchini and Capobianco, 2001), and at the same time, activate transcription factors related to luminal differentiation, as exemplified by the observed early activation of *Jun* or *Foxi1*, or recruit chromatin modifiers that could potentially tilt the balance of activation/repression on bivalent lineage promoters.

We report here that ectopic Notch activation results in the reprogramming of BCs into HR[neg] LCs. Given that in vivo Notch activation in BCs is mosaic, we do not observe an overt phenotype at the tissue level, since WT BCs that escaped tamoxifen induction can compensate for the mutant BCs that are lost to give rise to LCs. However, in organoids, we could document the clonal expansion of mutant N1ICD-expressing LCs, that appear to gain a competitive advantage and occasionally can form organoids composed exclusively of mutant nGFP[pos] luminal cells (Fig. 4A, lower row). Even if these mutant cells cluster close to WT LCs by UMAP analysis (Fig. 2A), it is well established that Notch gain-of-function mice can form mammary tumors (Bouras et al, 2008; Callahan and Smith, 2000; Diévart et al, 1999). Indeed, deregulated Notch activation has been shown to induce mammary carcinomas (Diévart et al, 1999) and to affect human mammary cell transformation (Stylianou et al, 2006), stem cell maintenance (Harrison et al, 2010) and be associated with poor outcome in breast cancer patients (Reedijk et al, 2005). Moreover, we found that constitutive Notch1 activation both when targeted to all mammary cells (with MMTV-Cre) and when restricted to HR[neg] (with N1Cre[ERT2] mice) results in pregnancy-dependent mammary hyperplasia (our unpublished observations). Of interest, our analyses suggest that the luminal HR[neg] cells generated by ectopic Notch1 activation in BCs (with both SMACre[ERT2] and K5Cre[ERT2]) are also susceptible to transformation, and they promote the growth of hyperplastic lesions upon successive rounds of lactation and involution (Fig. EV2G,H). These results carry important implications in breast cancer, revealing that Notch signaling is not only required for specifying luminal progenitor cells in the normal mammary gland, but that sustained and/or aberrant Notch activation in differentiated and lineage-committed cells has the potential to promote the appearance of mammary tumors, given its paramount role in the control of the delicate equilibrium between differentiation and proliferation required for tissue homeostasis. These observations reinforce the concept that the mechanistic processes through which stem cells commit to a particular differentiation path mirror those hijacked by oncogenes to trigger cellular transformation across various tissues (Blanpain and Fuchs, 2014). Therefore, unraveling these mechanisms is crucial for a better understanding of the genesis of cancer.

# Methods

## Reagents and tools table

| Reagent/resource | Reference or source | Identifier or catalog number |
|---|---|---|
| **Antibodies** | | |
| Rat anti-Keratin 8 | Developmental Studies Hybridoma Bank, University of Iowa | Cat# TROMA-I, RRID: AB_531826 |
| Chicken anti-Keratin 5 | Biolegend | Cat # 905901, RRID:AB_2565054 |
| Chicken anti-Keratin 14 | Biolegend | Cat # 906004, RRID:AB_2616962 |
| Alexa Fluor(R) 647 anti-mouse CD326 (EpCAM) | Biolegend | Cat # 118211, RRID:AB_1134104 |
| Anti-αSMA coupled with FITC | clone 1A4, F3777, Sigma-Aldrich | Cat # F3777, RRID:AB_3099704 |
| Rabbit Anti-GFP | Institut Curie antibody platform | |
| Cy5-conjugated anti-rat IgG | Invitrogen | A102525 |
| Cy3-conjugated anti-rabbit IgG | Invitrogen | A10520 |
| AlexaFluor 488 anti-chicken IgG | Invitrogen | A11039 |
| Alexa Fluor 488-conjugated anti-rabbit IgG | Invitrogen | A21206 |
| PE anti-mouse Epcam | Biolegend | Cat# 118206, RRID:AB_1134176 |
| APC/Cy7 anti-mouse CD49f | Biolegend | Cat# 313628; RRID:AB_2616784 |
| APC anti-mouse CD31 | Biolegend | Cat# 102510; RRID:AB_312905 |
| APC anti-mouse Ter119 | Biolegend | Cat# 116212; RRID:AB_313713 |
| APC anti-mouse CD45 | Biolegend | Cat# 103112, RRID:AB_312977 |
| **Chemicals, enzymes, and other reagents** | | |
| Triton X-100 | Euromedex | 2000-C |
| Paraformaldehyde | Electron Microscopy Sciences | 15710 |
| Sucrose | Sigma | S0389 |
| DMEM/F-12 | Gibco-Thermo Fisher Scientific | 21331020 |
| ITS | | |
| Collagenase A | Roche | 10103586001 |
| Hyaluronidase | Sigma-Aldrich | H3884 |
| DNAse I | Sigma-Aldrich | D4527 |
| Aqua-Polymount | Polysciences | 18606 |
| ProLong Diamond Antifade Mountant | Invitrogen-Thermo Fisher Scientific | P36930 |
| DMSO | Sigma-Aldrich | D2438 |
| Penicillin–streptomycin | Sigma-Aldrich | P4333 |

| Reagent/resource | Reference or source | Identifier or catalog number |
|---|---|---|
| Tissue-Tek O.C.T. | Sakura | 4583 |
| SUPERase-In RNase Inhibitor | Sigma-Aldrich | AM2694 |
| Corning® Matrigel® Matrix | Corning | 254234 |
| 4-Hydroxytamoxifen | Sigma-Aldrich | Cat #H6278 |
| Tamoxifen | MP Biomedicals | Cat #156738 |
| U0126 | Sigma-Aldrich | Cat #662005 |
| Aphidicolin | Sigma-Aldrich | Cat #178273 |
| **Software** | | |
| ImageJ | https://imagej.nih.gov/ij/index.html | |
| Rstudio | | |
| FlowJo | | |

## Methods and protocols

### Mice

SMACre$^{ERT2}$ (Wendling et al, 2009) and K5Cre$^{ERT2}$ (Indra et al, 1999) were crossed with a conditional gain-of-function Notch1 mutant mouse (Rosa-N1ICD-IRES-nGFP) (Murtaugh et al, 2003) or with the double fluorescent reporter Rosa26$^{mT/mG}$ (Muzumdar et al, 2007). Reporter expression was induced by intraperitoneal injection of tamoxifen (1 mg/10 g of weight) at postnatal day P21 for the analysis of mammary, salivary and lacrimal glands. For prostate analysis, adult males were induced with tamoxifen at 8 months of age and analyzed 7 weeks later. To assess tumor development, SMACre$^{ERT2}$/N1ICD females were induced at P21 and tumor were dissected after 1 or 2 rounds of pregnancy, lactation, and 10–15 days of involution.

### Ethics statement

All studies and procedures involving animals were in agreement with the recommendations of the European Community (2010/63/UE) for the protection of vertebrate animals used for experimental and other scientific purposes. Approval was provided by the ethics committee of the French Ministry of Research (reference APAFIS #34364-202112151422480). We comply with internationally established principles of replacement, reduction, and refinement in accordance with the Guide for the Care and Use of Laboratory Animals (NRC 2011). Husbandry, supply of animals, as well as maintenance and care in the Animal Facility of Institut Curie (facility license #C75–05–18) before and during experiments fully satisfied the animal's needs and welfare. All mice were housed and bred in a specific-pathogen-free (SPF) barrier facility with a 12:12 h light-dark cycle and food and water available *ad libitum*. Mice were sacrificed by cervical dislocation.

### Mammary gland dissociation and cell sorting

Mammary glands were harvested and digested with collagenase (Roche, 57981821, 3 mg/ml) and hyaluronidase (Sigma, H3884, 200 U/ml) for 90 min at 37 °C under agitation. Following washes, cells were dissociated with Trypsin for 1 min, dispase for 5 min (200 U/ml), and DNAseI (D4527, Sigma-Aldrich, 200 U/ml) and

then filtered through a 40 µm cell strainer to obtain a single cell preparation. Cells were incubated for 30 min with the following antibodies diluted at 1:100: APC anti-mouse CD45 (Biolegend), APC anti-mouse Ter119 (Biolegend), APC anti-mouse CD31 (Biolegend), PE anti-mouse Epcam (Biolegend), APC-Cy7 anti-mouse CD49f (Biolegend). Single-cell preparation was resuspended in flow buffer containing PBS, EDTA 5 mM, BSA 1%, FBS 1%, and DAPI. Dead cells (DAPI$^{pos}$) and Lin$^{pos}$ non-epithelial cells were excluded before analysis using the FACS ARIA flow cytometer (BD). The results were analyzed using FlowJo software. Single sorted cells were deposited in 96-well plates containing SUPERase-In RNase Inhibitor (20U/µl, Sigma, AM2694), 10% Triton X-10, and DEPC-treated H$_2$0 to library preparation using Smart-Seq2 protocol.

### Immunofluorescence on frozen sections

Mammary, salivary and lacrimal glands, and prostates were harvested and fixed at room temperature in PFA 4% for 1 h. Tissues were incubated for 3 days at 4 °C in sucrose 30% and embedded in optimal cutting temperature (OCT). Immunostainings were performed onto 10 µm sections. Antibodies used were rabbit anti-GFP (Institut Curie antibody platform, 1/300e), rat anti-K8 (TROMA-1, DSHB, 1/300e), chicken anti-K14 (906004, Biolegend, 1/500e), mouse anti-αSMA coupled with AF488 (clone 1A4, F3777, Sigma-Aldrich) and chicken anti-K5 (905901, BioLegend). Fluorochrome-conjugated secondary antibodies included Cy5-conjugated anti-rat IgG (A102525, Invitrogen), Cy3-conjugated anti-rabbit IgG (A10520, Invitrogen), Alexa-Fluor 488 anti-chicken IgG (A11039, Invitrogen), and Alexa Fluor 488-conjugated anti-rabbit IgG (A21206, Invitrogen).

### Organoid culture

Primary organoids were derived from 2- to 3-month-old female mice. Salivary, mammary and lacrimal glands, and prostates were dissected, pooled, and chopped to ~1-mm³ pieces and subjected to enzymatic digestion with 2 mg/mL collagenase A and 2 mg/mL trypsin for 30 min at 37 °C under agitation. Then, pieces were exposed to five rounds of differential centrifugation at 500×*g* for 15 s in order to remove stromal cells. The organoids were resuspended in DMEM/F-12 supplemented with 1× insulin–transferrin–selenium, 100 U/mL of penicillin, and 100 µg/mL of streptomycin.

Organoids were resuspended in Matrigel® (Corning) and plated at a density of 200 organoids/30 µl of Matrigel in 24-well plates. Matrigel drops were covered in a culture medium and incubated at 37 °C with 5% CO2. N1CD expression was triggered by 4-OHT (200 nM) in the culture medium for 24 h. To block proliferation, U0126 (5 µM) or Aphidicolin (0.6 µM) were added to the culture medium for 6 to 10 days. The medium was replaced every two days.

### Organoid staining

For immunostaining, organoids were fixed in 4% PFA for 10 min at room temperature, followed by 1 h of permeabilization (1% Triton in PBS) and 2 h incubation with blocking buffer (0.25% Triton/2% BSA/5% FBS/PBS). Primary antibodies were incubated overnight at 4 °C and secondary antibodies and DAPI for 5 h at room temperature. Antibodies used were rat anti-K8 (TROMA-1), rabbit anti-αSMA (NB600-531, Novus Biologicals), mouse anti-α-SMA coupled to FITC (clone 1A4, F3777, Sigma-Aldrich), rabbit anti-GFP (Institut Curie antibody platform), rabbit anti-Hes1

(11988, Cell Signaling), chicken anti-K5 (Biolegend, 905901) and secondary antibodies Cy3-conjugated anti-rabbit IgG (A102520, Invitrogen), Cy5-conjugated anti-rabbit IgG (A10523, Invitrogen), AlexaFluor 488 anti-chicken IgG (A11039, Invitrogen), and Cy5-conjugated anti-rat IgG (A10525, Invitrogen).

### Microscopy and image acquisition

For image acquisition of stained sections, a laser scanning confocal microscope (LSM780 or LSM880, Carl Zeiss) was used, equipped with a 40x/1.3 oil DICII PL APO objective. For image acquisition of organoids, we used an inverted spinning disk wide confocal microscope (CSU-W1, Nikon) equipped with a 40x/1.15 CFI APO LWD water objective.

## Single cell RNA-seq analysis

### Initial mapping, QC, and raw counts

Three batches of cells were processed and sequenced using Smart-seq2. FASTQ files were mapped to GRCm38 (mm10) using the STAR aligner (v2.7.7a) (Dobin et al, 2013). Downstream processing was performed using HTSeq (0.13.5) (Anders et al, 2015) to generate the raw gene count matrices.

Single cell gene counts and metadata were imported and analyzed in R (v4.3.0) using Seurat (4.3.0.1) (Butler et al, 2018; Satija et al, 2015; Stuart et al, 2019). Three batches of single cell RNA-Seq data were first processed individually, and then integrated. In batch 2, we filtered out cells from plate 5, since these cells were repeated from Batch1. ENSEMBL gene ids were converted to gene symbols using biomart (v2.56.1), using mouse genome annotation GRCm38, using the following settings: (biomart = 'genes', dataset = 'mmusculus_gene_ensembl', version = 102) (Durinck et al, 2009). Further analysis was carried out using the Seurat package.

First, features which were present in >2 cells, and cells having >200 features were selected for analysis. Next, filtering was performed to retain cells with <10%mitochondrial reads, >1400 features (genes), and an RNA count of >100,000. The dataset was normalized using Log normalization. Gene expression counts were scaled for all genes. Next, principal component analysis (PCA) was performed using the 2000 most highly variable genes. Clustering was performed using the first 30 principal components (PCs), using a resolution of 0.5. UMAP was created using the first 30 PCs, using a spread of 0.4.

Next, non-mammary-epithelial cell types were filtered. Stromal and salivary cells (cells with gene counts of Epcam <2, Dcpp1 >1, respectively), were filtered out. Data normalization, PCA, clustering and UMAP steps above were repeated after cell filtering.

Afterward, integration of all three batches was performed using Seurat. During integration, batch 1 was used as a reference, since it had a balanced representation of basal, intermediate, and luminal cell types (determined previously using FACS). After integration, the dataset was scaled, and PCA, clustering and UMAP steps were carried out once again on the integrated dataset, using a clustering resolution of 0.5 and UMAP spread of 0.4.

Selection of this cluster resolution was based on the stability of clusters (using tool clustree) (Zappia and Oshlack, 2018), gene expression patterns after clustering (using Seurat's findMarkers), and FACS cell type labels. Four clusters were noted. The gene expression markers and FACS labels within clusters were examined. Cluster 1 was composed of basal and intermediate cells

(labeled by FACS), hence this cluster was further split (clustering resolution 0.6), giving 2 subclusters: one having a more basal gene expression pattern, and another having a less basal, more intermediate gene expression. Based on the expression of known cell type markers and FACS labels, we labeled the five resulting clusters as BAS, INT1, INT2, HR$^{pos}$, and HR$^{neg}$. Markers for these clusters were determined using Seurat's FindAllMarkers tool, using the settings only.pos=TRUE (selecting positive markers), min.pct = 0.40 (only test markers which are expressed in at least 40% of cells), and test.use = "roc". Markers were ordered by log$_2$ fold change, and the top seven markers for each cluster were plotted in a heatmap.

To estimate and visualize the expression of basal and luminal gene signatures in each cell, two approaches were used. In the first approach, Seurat's AddModuleScore function was employed, using basal and luminal signatures from Kendrick et al (Kendrick et al, 2008). Additionally, Luminal ER positive and ER negative signatures were combined to generate a common luminal score.

As an alternative method to visualize gene expression changes during the basal/ luminal transition, we used an adjusted proportion score-based approach. For this, we first employed Seurat's FindMarkers (test = "roc") to find the top 50 gene markers (based on AUC score) expressed in our BAS and HR$^{neg}$ clusters. Using these signatures, we followed a similar approach to the one reported in Wuidart et al (2018). Briefly, in each cell, we counted the number of BAS marker genes expressed above a threshold (specifically, the median expression of BAS genes), and then we divided this value by the total number of BAS markers, giving the proportion of BAS markers expressed. Since this proportion score is also affected by the total number of genes expressed in each cell, we corrected for this effect using a linear modeling approach, by means of the rlm function from the MASS package in R. This gave us the corrected, adjusted proportion BAS score. The same approach was then applied with the HR$^{neg}$ luminal markers, generating the adjusted proportion HR$^{neg}$ luminal score.

Seurat's AddModuleScore function was also used to calculate cell cycle module scores, using genes from GO and KEGG (GO positive regulation of cell cycle: GO0045787, KEGG_CELL_CY-CLE.v2023 from Msigdb). To evaluate G1 and G2/M scores, we used Seurat's CellCycleScoring function. To study genes differentially expressed in the proliferative subgroup, we used Seurats' CellSelector tool on the UMAP to manually define the group of cells showing a high cell cycle score, and used FindMarkers to find the genes most significantly enriched in this group.

### Label transfer

Comparison of the Smart-N1ICD with other datasets was performed using label transfer in Seurat. This process allows the classification of cells in a query dataset, using another dataset as a reference. Corresponding cell type labels from the reference dataset are transferred to the query dataset, as a 'predicted ID'. Original clusters (from the query dataset) and corresponding predicted IDs (from the reference dataset) were compared using alluvial plots, by ggplot (https://ggplot2.tidyverse.org), ggalluvial (v0.12.5) (http://corybrunson.github.io/ggalluvial/) (Brunson, 2020). Using this approach, we compared our dataset to datasets from Wuidart et al (Wuidart et al, 2018), Centonze et al (Centonze et al, 2020), Pal et al (Pal et al, 2021), and Bach et al (Bach et al, 2017). We used the same QC filtering criteria described in these publications, with

the following changes, to aid comparison with the Smart-N1ICD dataset: the analysis was performed using Seurat, and log normalization was used. Data were scaled, PCA, clustering and UMAP steps were run, using the same settings as for our Smart-N1ICD dataset. For Wuidart et al, gene expression clusters were defined as BAS, HR[neg], HR[pos], and EMP. For Centonze et al, after clustering, the resulting cell subtypes were re-labeled as BAS, HR[neg], HR[pos], and hybrid. For Pal et al, clusters from the TEBs and Duct datasets were labeled as BAS, HR[pos], and HR[neg]. For Bach et al, we used cluster labels as defined in their shared metadata file.

### SCENIC analysis

To infer transcription factor (TF) activity, SCENIC analysis was performed using pySCENIC (0.12.1) (Aibar et al, 2017; Van de Sande et al, 2020), using default parameters. Smart-seq2 gene expression counts and metadata were imported to create Anndata files (Scanpy, v1.7.2), quality checks were performed, and cells were filtered using the same parameters as described for Seurat analysis, to exclude all low-quality cells. All three batches were then concatenated and converted to a loom format for analysis with the pySCENIC pipeline. First, genes correlating with TFs expression were inferred using the GRN step, resulting in TF modules. Next, the CTX step was used to prune genes from these modules, to retain only genes which contain the associated TF motif within cis-regulatory regions. These pruned modules represent the regulons of TFs and their associated downstream targets. Lastly, the activity of the regulons was estimated as an area under curve (AUC) in the AUCell step of the analysis. From the resulting loom file, a matrix of AUC values was extracted and then imported into an R environment for further analysis. The AUC matrix and the Smrt-NIC Seurat object were both filtered to contain the same set of cells. Next, cluster annotation labels (derived from the Seurat object) were used to perform Regulon specificity score (RSS) analysis on the SCENIC AUC matrix, to infer which regulons show the strongest cluster-specific activity. The activity of selected regulons was visualized with UMAP plots.

### Trajectory analysis

Trajectory analysis was performed on the Smart-N1ICD Seurat object using slingshot (v2.8.0) (Street et al, 2018). Slingshot analysis was performed on the PCA structure, specifying the BAS cluster as the origin. Principal curves were plotted and two trajectories were observed.

Next, genes whose expression correlated with the trajectory #1 (from BAS to HR[neg]) pseudotime were inferred using tradeseq (v1.14.0) (Van den Berge et al, 2020), which uses a generalized additive model (GAM) to fit the variation of expression of each gene along pseudotime. The optimal number of knots was estimated for our dataset at 8. This analysis was performed on the 4000 most highly variable genes in the dataset. Next, an association test was performed to identify genes correlating with pseudotime, using settings lineages = TRUE and contrastType = " consecutive". The genes were ordered based on the statistical significance of this correlation (Wald statistic). We visualized the top 40 genes significantly associated with trajectory #1, as well as the complete list of genes significantly associated with this trajectory (filtering out low expression genes—selecting genes having a count of at least 1 in at least 20 cells, giving 357 genes). To visualize these genes, their integrated expression values were extracted from the Seurat object, ordering cells along trajectory #1 pseudotime. Gene expression values in the resulting matrix were smoothed using a binning/rolling window process along

pseudotime, using the rollapply function from the "zoo" package (1.8–12) (Zeileis and Grothendieck, 2005). Correspondingly, a similar bin smoothing was applied to cluster annotations, selecting the most common cluster annotation within each bin. The resulting matrix was plotted as a heatmap using the pheatmap package (v1.0.12) (https://cran.r-project.org/web/packages/pheatmap/index.html), along with cluster annotations, using settings cutree_rows = 3 (for the top 40 genes), and cutree_rows = 6 (for all trajectory associated genes). Clustering_method = "average" and package viridis (v0.6.3) (Garnier et al, 2024) were used for the heatmap color palette. For selected genes, integrated gene expression was plotted in single cells ordered along trajectory #2 in pseudotime. For GO analysis of genes varying along the trajectory, we used the compareCluster function from the clusterprofiler package (Yu et al, 2012).

### Statistical tools

Analysis in R (4.3.0) was performed within RStudio (2023.06.0 + 421). Analysis in Python (SCENIC) was performed using Python (3.10.4) within a conda environment (4.7.12), using Jupyter notebook (6.4.10). ggplot-based plots were created with ggplot2 (v3.4.2).

### Statistics and reproducibility

Animals were randomized and analyzed in a non-blinded manner. All graphs show mean ± SEM.

For each experiment, at least $n = 3$ biological replicates were analyzed. Graphs and statistical analyses were performed using GraphPad Prism (v 10.2.2).

## Data availability

Smart-Seq2 scRNA-sequencing data generated in this study is accessible on the Gene Expression Omnibus GEO repository (GSE268822). All source data containing the raw images have been submitted to BioImage Archive and have been assigned the BioImages accession number S-BIAD1634. Analysis codes are available upon request.

The source data of this paper are collected in the following database record: biostudies:S-SCDT-10_1038-S44318-025-00424-1.

## Peer review information

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

## Acknowledgements

We acknowledge Sarah Kinston and Berthold Göttgens at Wellcome Trust-MRC Cambridge Stem Cell Institute, University of Cambridge, UK, for performing the Smart-seq2V2 scRNA-sequencing experiments. The authors also wish to warmly acknowledge the flow cytometry and cell sorting platform at Institute Curie for their technical support, the in vivo experimental facility for help in the maintenance and care of our mouse colony, and the Cell and Tissue Imaging Platform PICT-IBiSA at Institut Curie, member of the national infrastructure France-Bioimaging (https://ror.org/01y7vt929), supported by the French National Research Agency (ANR-24-INBS-0005 FBI BIOGEN) and the Cell and Tissue Imaging (PICT-IBiSA) of the Genetics and Developmental Biology Unit (UMR3215/U934) for their expertise and assistance. We are very grateful to all members of the Fre laboratory for their support, critical reading of the manuscript, and constructive discussions. This work was funded by Paris Sciences et Lettres (PSL* Research University) (grant # C19-64-2019-228), the French National Research Agency (ANR) grant numbers ANR-21-CE13-0047 and ANR-22-CE13-0009, the Medical Research Foundation FRM "FRM Equipes" EQU201903007821, the FSER (Fondation Schlumberger pour l'éducation et la recherche) FSER20200211117, the Association for Research against Cancer (ARC) label ARCPGA2021120004232_4874, the Worldwide Cancer Research Foundation # 24-0216 and by Labex DEEP ANR-Number 11-LBX-0044 to SF. SR received funding from the European Research Council (ERC, grant agreement no. 950349). CM was financed by a post-doctoral fellowship from ARC (ARCPDF12021020003033). The funders had no role in study design, data collection and analysis, decision to publish, or preparation of the manuscript.

## Author contributions

**Candice Merle**: Conceptualization; Formal analysis; Supervision; Funding acquisition; Validation; Investigation; Methodology; Writing—original draft; Writing—review and editing. **Calvin Rodrigues**: Data curation; Formal analysis; Methodology. **Atefeh Pourkhalili Langeroudi**: Investigation; Methodology. **Robin Journot**: Methodology. **Fabian Rost**: Data curation. **Yiteng Dang**: Data curation. **Steffen Rulands**: Data curation; Project administration. **Silvia Fre**: Conceptualization; Funding acquisition; Writing—original draft; Project administration; Writing—review and editing.

Source data underlying figure panels in this paper may have individual authorship assigned. Where available, figure panel/source data authorship is listed in the following database record: biostudies:S-SCDT-10_1038-S44318-025-00424-1.

## Disclosure and competing interests statement

The authors declare no competing interests.

# Expanded View Figures

**Figure EV1.  Transcriptional signatures characterizing the different cell clusters identified by UMAP analysis.**

Related to Fig. 2. (**A**) Representative images of SMACre^ERT2/mTmG mammary gland (MG), lacrimal gland (LG), salivary gland (SG), and K5Cre^ERT2/mTmG prostate sections induced at P21 and analyzed 6 weeks later by immunofluorescence for mGFP (yellow) and the basal marker α-SMA (white), demonstrating that α-SMA^pos cells are exclusively BCs, indicating unipotency. Scale bars represent 50 μm. (**B**) Gating strategy used for cell sorting experiments. Doublets, Lin^pos and dead cells (DAPI^pos) were excluded from further analysis. (**C**) FACS plots showing the gates defining luminal (EpCAM^high/Cd49f^low), intermediate (EpCAM^med/Cd49f^med), and basal (EpCAM^low/Cd49f^high) sorted cells from SMACre^ERT2/N1ICD mammary glands induced at P21 and chased for 1, 3, or 6 weeks, that were used for SMARTseqV2. The percentages of each population are indicated for one representative experiment. (**D**) UMAP plot showing the distribution of each sequenced cell in the different clusters based on the Cre mouse used to target them (K5Cre^ERT2 or SMACre^ERT2). (**E**) UMAP plots show the expression of well-defined markers for basal (*Krt5, Krt14*), luminal (*Krt8, Krt19*), and HR^pos (*Esr1, Pgr*) cells. (**F**) Violin plots representing basal and luminal scores, based on signatures from (Kendrick et al, 2008) for each cluster. (**G**) UMAP plot indicating the cell cycle phases for each sequenced cell. The boxed cells correspond to the proliferative population highlighted in Fig. 2E. (**H**) UMAP plots are color-coded according to the expression of the single-cell G2M and S score. (**I**) Heatmap showing the genes presenting a high expression in the proliferative group. (**J**) Proportion of cells in different cell cycle phases (G1, G2M, or S) in each cluster.

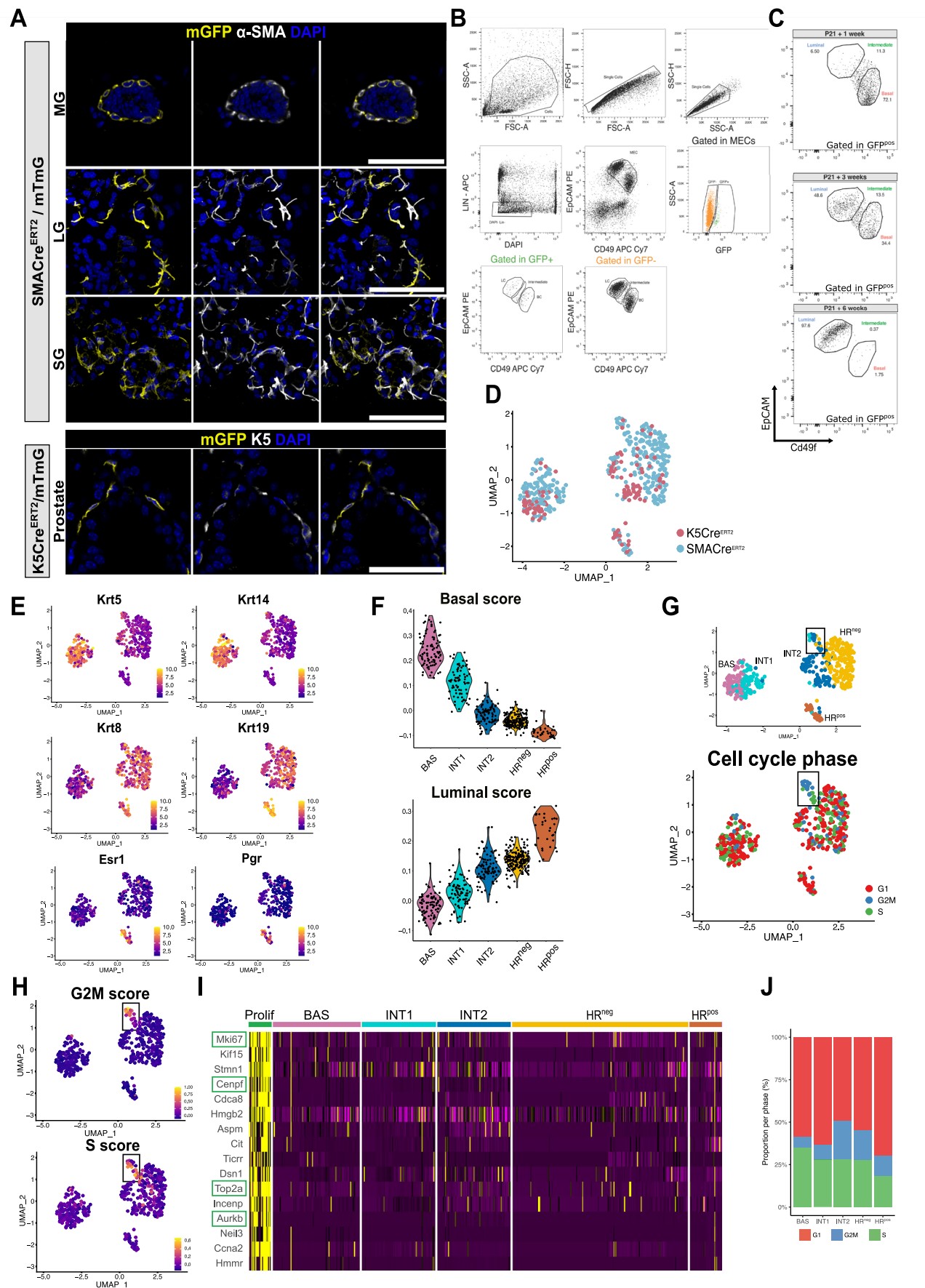

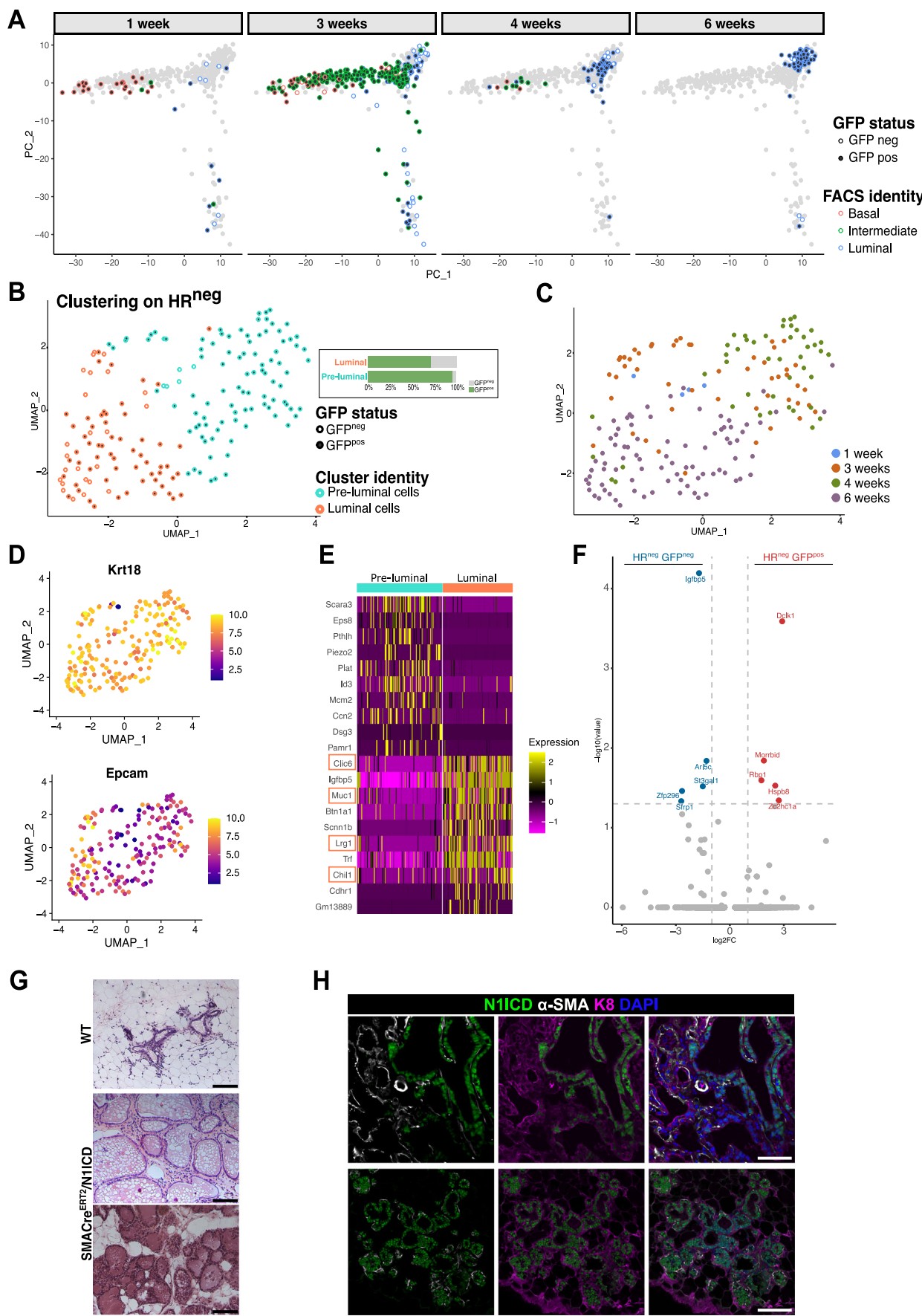

◀ **Figure EV2. Lineage trajectories of cell fate switch and associated transcriptional features.**

Related to Fig. 3. **(A)** PCA plots highlighting changes in cell transcriptional state along the basal-luminal differentiation trajectory, at different timepoints following N1ICD activation. Colored dots indicate the FACS gate information for each index-sorted cell, based on the cell surface markers EpCAM and Cd49f. **(B)** UMAP refined analysis of the HR[neg] cluster indicating GFP[pos] and GFP[neg] cells. The two new resulting subclusters appearing are indicated by different colors (turquoise and orange). The boxed graph represents the percentage of GFP[pos] (green) and GFP[neg] (gray) cells for each cluster, showing the predominance of GFP[pos] mutant cells in the pre-luminal cluster. **(C)** UMAP representation of the distribution of mutant nGFP[pos] cells belonging to the HR[neg] cluster color-coded based on the different timepoints after N1ICD induction, as indicated. **(D)** UMAP representations of the two luminal subclusters show the expression levels of two luminal-specific genes (*Krt18* and *Epcam*) for each cell. **(E)** Heatmap showing differentially expressed genes (DEGs) distinguishing the pre-luminal and luminal clusters. **(F)** Volcano Plot of DEGs between HR[neg] GFP[pos] (170 cells) (red dots) and HR[neg] GFP[neg] (34 cells) (blue dots) cells. **(G)** Hematoxylin and Eosin staining of WT (wild-type) mammary gland and mammary tumor sections following N1ICD activation. Wild-type mouse mammary glands were collected after one pregnancy, while tumors were harvested after three pregnancies, followed by 10 to 15 days of involution. **(H)** Immunofluorescence anti-GFP (corresponding to N1ICD), α-SMA, and K8 on sections of mammary tumors developed upon N1ICD activation, three pregnancies, and 10 days of involution showing the clonal expansion of mutant nGFP[pos] cells. Nuclei are stained with DAPI. Scale bars represent 100 μm in **(G, H)**.

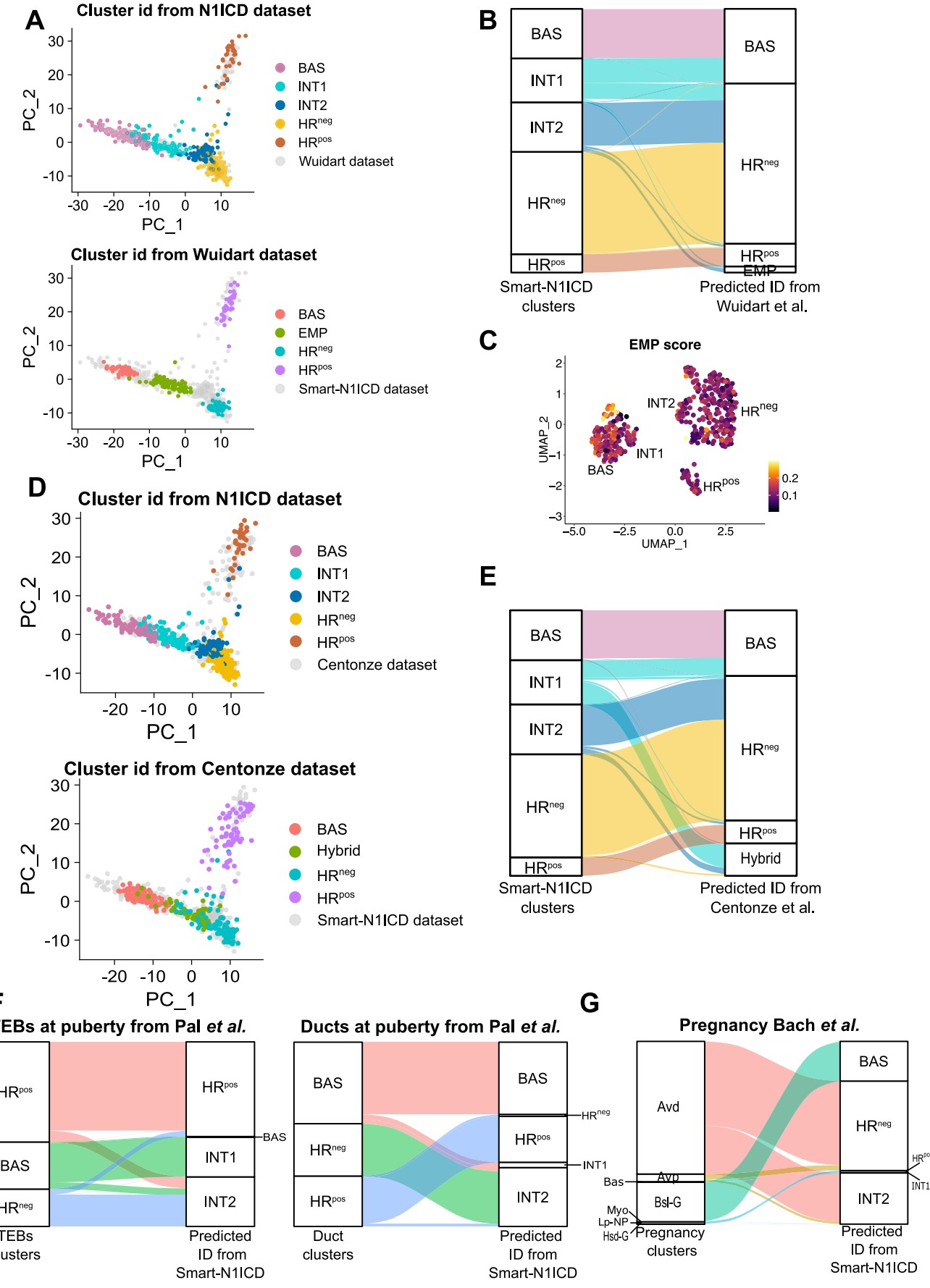

**Figure EV3. Integration of SMART-N1ICD dataset with scRNAseq from embryonic and adult hybrid mammary cells.**

Related to Fig. 3. **(A)** PCA plots showing the integration of our data (SMART-N1ICD) and the dataset from Wuidart et al (Wuidart et al, 2018). **(B)** Alluvium plots showing label transfer of each cell cluster using the Wuidart dataset as reference. **(C)** UMAP plot representing enrichment for the EMP score derived from Wuidart et al, dataset. Purple represents the lowest EMP score and yellow the highest EMPs score. **(D)** PCA plots showing the integration of our data (SMART-N1ICD) and the dataset from Centonze et al, (Centonze et al, 2020). **(E)** Alluvium plots showing label transfer of each cell cluster using the Centonze dataset as reference. **(F)** Alluvium plots showing label transfer of each cell cluster in the Pal et al, dataset either from TEBs (left) or from pubertal ducts (right) using our dataset (Smart-N1ICD) as reference. **(G)** Alluvium plot showing label transfer of each cell cluster in the Bach et al., dataset performed at pregnancy using our dataset (Smart-N1ICD) as reference.

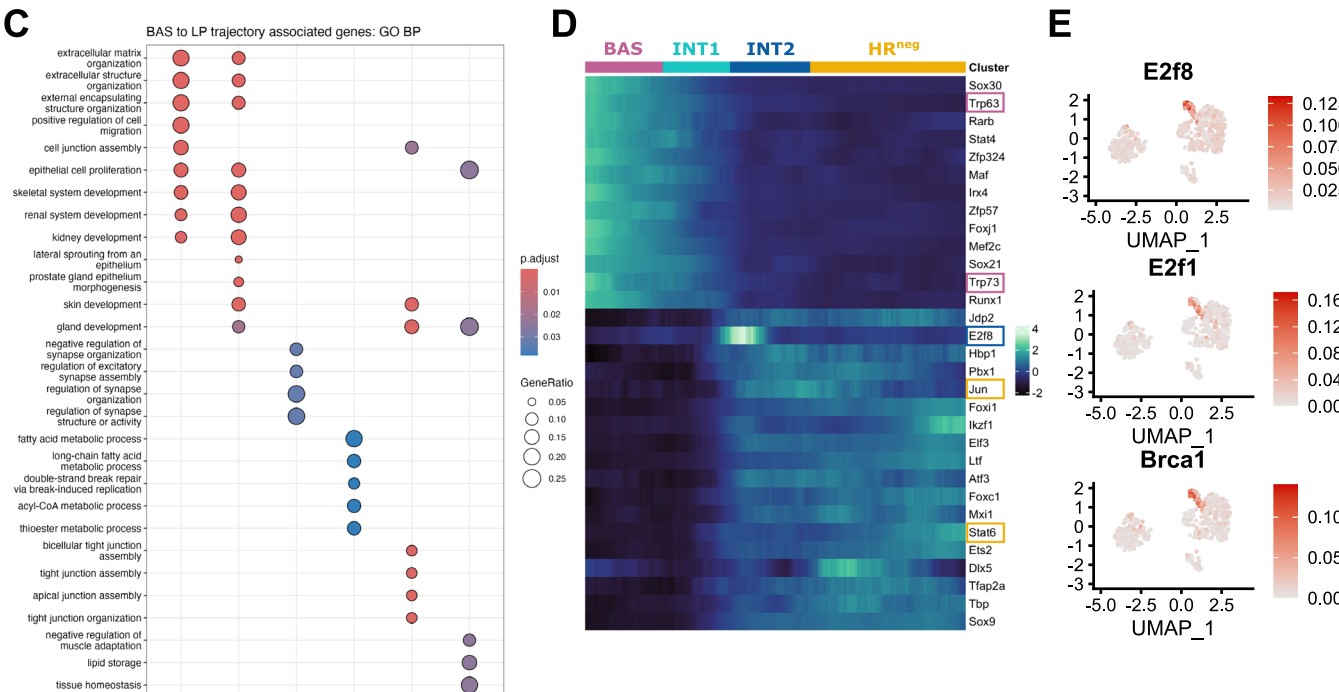

◀ **Figure EV4.** **Pseudotime ordering identifies the transcriptional signatures associated with the progressive lineage transition from BCs to LCs.**

Related to Fig. 3. (**A**) Heatmap illustrating the top 40 genes exhibiting a differential pattern of expression along the pseudotime from basal to luminal (HR^neg) identity. The clusters are color-coded, as in Fig. 2B. (**B**) Heatmap showing the dynamic expression profile of each gene towards the lineage switch trajectory, distinguishing six different patterns of expression along the process of basal to luminal transition. (**C**) GO terms associated with the genes defining the six groups in (**B**). *p* values were defined using one-sided Fisher's exact statistical test. (**D**) Heatmap showing the top transcriptional regulons specific to each cluster (based on RSS analysis), plotted along the pseudotime trajectory 1. Boxed genes are described in the text. (**E**) UMAP plots showing the expression of regulons specific to cluster INT2, based on Fig. 3E.

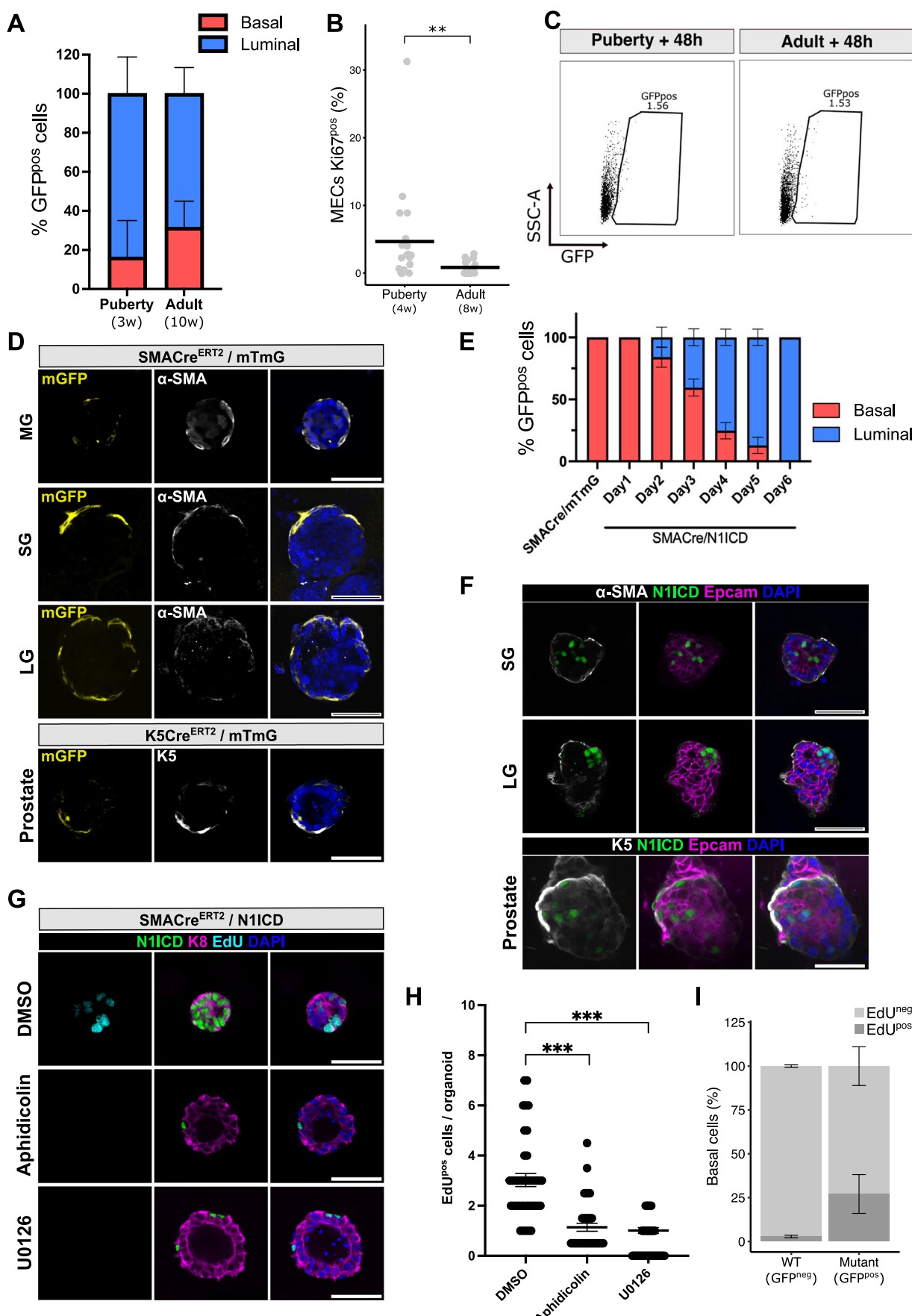

◄ **Figure EV5. Lineage switch and BCs unipotency are recapitulated in organoids.**

Related to Figs. 4, 5. (**A**) Quantification of the proportion of GFP$^{pos}$ cells in basal and luminal FACS gates after induction at pre-puberty (3w) or adulthood (10w), showing the incomplete cell fate switch after a 6-week chase in adult mice. Data were displayed as mean ± SEM, and represent at least three mice. (**B**) Percentage of Ki67-positive mammary epithelial cells (MECs) in pubertal (4w) or adult (8w) mice. $n = 3$ mice with a total of 16 sections for pubertal mice and 19 sections for adult mice. $p$ value $= 0.009$ defined by $t$-test. (**C**) FACS plots showing the percentage of mutant GFP$^{pos}$ cells after 48 h induction in pubertal or adult mice, demonstrating invariant recombination efficiency. (**D**) Representative immunofluorescence images showing mGFP, α-SMA, and K8 expression, 6 days after 4-OHT induction of SMACre$^{ERT2}$/ mTmG control organoids. Scale bar 50 µm. (**E**) Quantification of the proportion of mutant nGFP$^{pos}$ basal and luminal cells per organoid at the indicated times after induction in SMACre$^{ERT2}$/N1ICD mice or 6 days after tamoxifen in SMACre$^{ERT2}$/mTmG control animals. Error bars represent mean ± SEM. A minimum of 18 organoids were analyzed for each condition and included three independent experiments. (**F**) Representative images of SMACre$^{ERT2}$/N1ICD (SG and LG) or K5Cre$^{ERT2}$/ N1ICD (prostate) organoids showing mutant cells (nGFP$^{pos}$), featuring the expression of the luminal marker Epcam and the absence of basal markers (α-SMA or K5, as indicated). Scale bar 50 µm. (**G**) Representative images of EdU staining in SMACre$^{ERT2}$/N1ICD organoids treated with DMSO, Aphidicolin, and U0126. Scale bar represents 50 µm. (**H**) Quantification of EdU$^{pos}$ cells per organoids after 6 days in culture with DMSO, Aphidicolin, or U0126. *** indicates $p$ value <0.0001 ($p$ value $= 2.04$e-10 for DMSO/ Aphidicolin and 2.04e-10 for DMSO/U0126, using Wilcoxon test). Data were displayed as mean ± SEM and represented three independent experiments with at least 40 organoids in total. (**I**) Quantification of the proportion of EdU positive or negative WT (GFP$^{neg}$) or mutant (GFP$^{pos}$) BCs in organoids 3 days after induction. Data were displayed as mean ± SEM and 19 organoids were analyzed from three independent experiments.

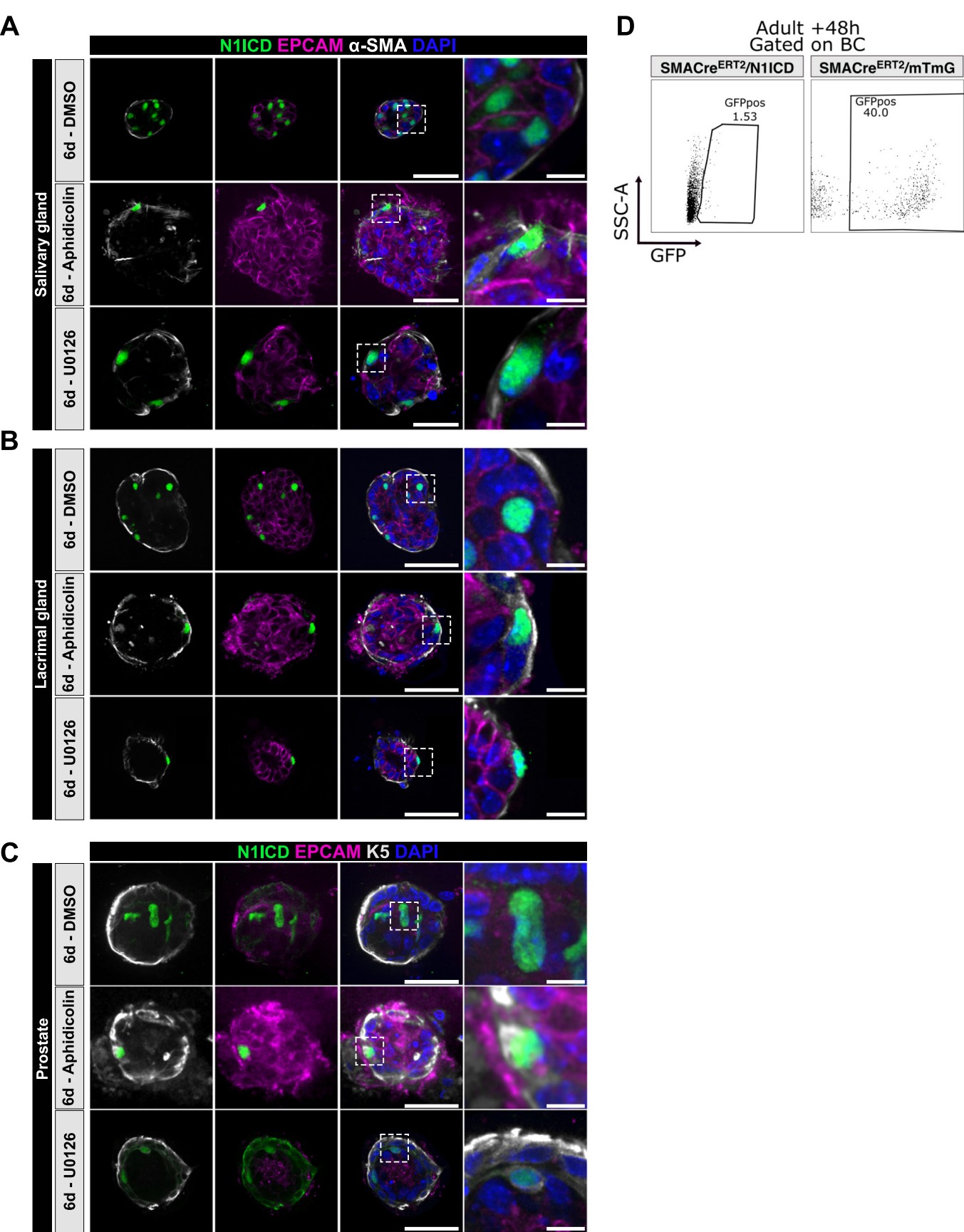

◄ **Figure EV6.** **Proliferation is an obligatory step for lineage transition in salivary, lacrimal, and prostate organoids.**

Related to Fig. 4. (**A–C**) Representative images showing immunofluorescence for nGFP (N1ICD in green), the luminal marker Epcam (purple) and the basal gene α-SMA (white) in SMACre[ERT2]/N1ICD salivary (**A**) and lacrimal (**B**) gland organoids, or the basal gene K5 (white) in K5Cre[ERT2]/N1ICD prostate (**C**) organoids treated with DMSO, Aphidicolin or U0126 for 6 days. Nuclei are stained with DAPI in blue. The scale bar represents 50 μm in (A–C) and 10 μm in the magnified insets. (**D**) Quantification of the proportion of basal GFP[pos] cells after 48 h of induction in SMACre[ERT2]/N1ICD and SMACre[ERT2]/mTmG adult mice, showing the difference in recombination efficiency with the two Cre drivers.

