## [Peer Review File · The EMBO Journal]

Transcriptional landscapes underlying Notch-induced lineage conversion and plasticity of mammary basal cells

Candice Merle, Calvin Rodrigues, Atefeh Pourkhalili Langeroudi, Robin Journot, Fabian Rost, Yiteng Dang, Steffen Rulands, and Silvia Fre

Corresponding author: Silvia Fre (silvia.fre@curie.fr)

Review Timeline:

Transferred from Review Commons:	22nd Oct 24
Editorial Decision:	25th Oct 24
Revision Received:	6th Feb 25
Editorial Decision:	7th Mar 25
Revision Received:	12th Mar 25
Accepted:	19th Mar 25

Editor: Daniel Klimmeck

Transaction Report:

This manuscript was transferred to The EMBO Journal following peer review at Review Commons.

Review #1**1. Evidence, reproducibility and clarity:****Evidence, reproducibility and clarity (Required)**

In the current study, Fre and colleagues studied the mechanisms by which ectopic activation of Notch signaling in mammary basal cells promotes a stepwise transition from basal to luminal cell identity in adult mouse mammary glands. Also, they tested the conservation of this mechanism in other glandular epithelia such as the lacrimal gland, the salivary gland and the prostate. To uncover the molecular mechanism underlying this progressive switch, they performed single-cell transcriptomic analysis by SMART-Seq on index-sorted mutant mammary cells at different stages of lineage conversion coupled with assays in organoid cultures. They demonstrated that the proliferation of basal cells is required to convert them into luminal progenitors following NICD expression.

The manuscript is interesting and the data of high quality. The novelty of this manuscript relies in the use of single-cell sequencing to study the mechanisms of cell plasticity and the demonstration that this mechanism is conserved in other glandular epithelia such as lacrimal gland, salivary gland and prostate. The involvement of proliferation in this transition is a novel aspect compared to their previous study.

This manuscript will be interesting for those studying stem cell and development biology, plasticity, Notch signalling and fate conversion and will appeal the broad range of readers from EMBO J or Developmental Cell. We provided some suggestions to further improve the manuscript.

****Major comments:****

1. The data suggesting that ectopic Notch activation in basal cells of other epithelia should be strengthened and quantified. The immunofluorescence images shown are not always clear and informative. Quantification of cell types should be performed as they did for mammary glands. (Fig. 1 C, D, E)
2. The number of cells sequenced seems low. Could the authors explain why the number of retrieved cells is low and demonstrate that despite this low number of cells, the results obtained are reliable?
3. The results concerning proliferation linked to the intermediate clusters should be more detailed. The authors should provide more information regarding the high proliferative group within the INT2 cluster. Which genes relative to cell cycle progression are upregulated in this group? It would also be important to assess in which specific phases of the cell cycle are all the clusters. Is the proliferative group in S phase?

4. The difference between WT luminal cells and mutant cells should be strengthened. The authors should provide more robust information to exemplify the difference between WT luminal cells and mutant luminal cells. Indeed, the PCA plot reported in Fig. S3B is not really showing a clear separation between the 2 groups. Moreover, only one of the feature plots (Clic6) showed a clear difference between the 2 groups. The authors should check also other genes other than the luminal markers genes to identify the underlying differences between these luminal cells.
5. The authors should perform further analysis of the transcriptomic data and not only compared these data with the already published dataset. By doing so, the authors might be able to propose novel molecular mechanisms driving the cell fate switch upon Notch activation.
6. The author should provide more data regarding the difference observed between adult and pubertal mice. Which is the proportion of recombined cells at T0 (just after the induction)? The difference observed in the kinetic switch between puberty and adulthood, with the given data, can be explained by differences in the recombination proportion. Also, difference in proliferation of the cells between puberty and adulthood should be also investigated.
7. The authors should provide more precise information regarding the EdU quantification presented in Fig. S6 C,D. Are the mutant basal cells more proliferative than the WT? Quantifying the GFPpos/EdU pos basal cells would help answer this question.
8. The author should provide more information to sustain the following statement "Some mGFPpos cells instead remained in the basal compartment after mitosis, undoubtedly representing WT BCs that only floxed the mTmG reporter but not the more refractory N1ICD allele".
9. The statement relative to Notch activation leading to hyperplastic lesions, reported between line 423 and 429, should be strengthened by including data in the current manuscript.

****Minor comments****

1. Typos are found across the manuscript (missing space and missing bold font while citing figures)
2. The sentence reported between lines 88 and 93 is not of immediate understanding due to complicated syntax.
3. Relatively to Fig. 1B: can the authors explain why the quantification on the timepoint 2w results less than 100%?
4. Relatively to Fig. S1D: can the author explain why the gates shown in the FACS plots are different between different timepoints?

5. Relatively to Fig. 2C: authors should provide another graphical representation about the similarities between each identified cluster either to luminal or basal identity. We suggest plotting each cluster separately and drawing the diagonal in such a way to make the visualization easier.

6. Relatively to Fig. 4C: the graph presented is not clear. Is the graph reporting the percentage of GFP positive cells on the SMA positive cells or rather the percentage of SMA positive cells that are GFP positive?

More importantly, important quantitative data are missing in this graph. Indeed, no WT (no NOTCH activation) nor DMSO CTRL data were included in this graph.

7. Relative to Fig. 5C: the author should provide quantitative data to corroborate their observations.

2. Significance:

Significance (Required)

This manuscript will interest those studying stem cell and development biology, plasticity, Notch signaling, and fate conversion. It will appeal to a broad range of readers from EMBO J or Developmental Cell.

3. How much time do you estimate the authors will need to complete the suggested revisions:

Estimated time to Complete Revisions (Required)

(Decision Recommendation)

Between 1 and 3 months

4. Review Commons values the work of reviewers and encourages them to get credit for their work. Select 'Yes' below to register your reviewing activity at Web of Science Reviewer Recognition Service (formerly Publons); note that the content of your review will not be visible on Web of Science.

Yes

Review #2

1. Evidence, reproducibility and clarity:

Evidence, reproducibility and clarity (Required)

The manuscript titled "Transcriptional landscapes underlying Notch-induced lineage conversion and plasticity of mammary basal cells" by Candice Merle et al. investigates the mechanisms by which Notch signaling influences lineage conversion in mammary basal cells. The study utilizes single-cell transcriptomics and organoid cultures to explore the molecular pathways involved in this process and underscores the broader implications of these findings in other glandular epithelia. The research presents significant insights into cellular plasticity and the role of Notch signaling in cancer biology, yet some novelty is lacking with respect to previous findings regarding Notch activation in the cell state transition from basal to luminal fate. While the research is innovative and thorough, there are several areas that could benefit from further exploration to strengthen the findings and broaden the implications. Detailed characterization of intermediate cell states, temporal dynamics, and functional validation of key regulators are needed to enhance the robustness of the conclusions. Otherwise, this is a very thoughtful and insightful study.

2. Significance:

Significance (Required)

1. While the study identifies intermediate cell states between the basal and luminal fates, further characterization and functional assays (e.g., mammosphere, proliferative potential, differentiation potential) would help to clarify the roles and heterogeneity of these intermediate states.
 - a. It would be extremely insightful to investigate how these intermediate states contribute functionally to the lineage conversion process and their potential implications in the transition from basal to luminal states after Notch activation.
 - b. Have the authors attempted to perform loss-of-function studies for candidate genes (like *Tacc3* or *Racgap1*) required to mediate the Notch-mediated basal to luminal commitment?
 - c. Are any genes from the INT1 and INT2 clusters able to be validated in situ?
2. The addition of salivary, lacrimal, and/or prostate organoids would benefit the broader applicability of the proliferation requirement for this state transition.
3. In Figure 1 it would help to include the quantitative data for the other organs.
4. For the organoid studies in Figure 4, it would be helpful to include quantitation of the eGFP within basal and luminal states over time post NICD induction. Between day 3 and day 6 could capture more nuanced changes occurring within the transition, along with the inclusion of proliferation markers like BrdU in the analysis within these studies including the Aphidicolin and U0126 treatments.
5. Have the authors considered ChIP-seq or ATAC-seq to complement the scRNA sequencing studies that could add to the lineage trajectories identified?

3. How much time do you estimate the authors will need to complete the suggested revisions:

Estimated time to Complete Revisions (Required)

(Decision Recommendation)

Between 3 and 6 months

Yes

Review #3

1. Evidence, reproducibility and clarity:

Evidence, reproducibility and clarity (Required)

In the manuscript, "Transcriptional landscapes underlying Notch-induced lineage conversion and plasticity of mammary basal cells" by Merle et.al, the authors elucidate the how ectopic expression of N1ICD in basal mammary cells (and other epithelial) reprograms them to a luminal cell fate. Using single cell RNA-sequencing they find that the change is progressive from basal \diamond luminal and involves 2 intermediate stages labelled as INT1 and INT2. The authors also find that there are no unique genes expressed by these subsets, but they are similar in transcriptional space to other EMP/hybrid cells described previously. Interestingly, some of the INT2 cells show higher cell cycle score. Therefore, the authors test and conclude that proliferation of basal cells in addition to N1ICD expression is required for change to luminal identity consistent with previous studies.

Below are a few points to consider regarding the findings which in my opinion will clarify and enhance the findings of the paper.

1. Is the INT1/INT2 state present during pubertal development in TEBs or during pregnancy?

Can the authors perform a similar label transfer analysis using the single cell datasets generated from other studies at these developmental time points? (OPTIONAL but will be very interesting to do)

2. Have the authors considered using entropy-based tools to understand whether the INT1/2 state is indeed multipotent such as StemID or cellular potency tools such as CytoTRACE? (OPTIONAL but will be interesting to do)

3. The authors conclude that the basal cells need to proliferate to become luminal - however in the UMAPs on Fig 2E, most of the cells in the INT2 cluster don't have a high cell cycle score or expression of cell cycle related genes Fig S5C? Is this due to the timing of when the cells were sorted for sequencing?

4. Fig. S6C not all N1ICD+ cells are EdU+ - it would be great if authors can clarify what percent of N1ICD expressing basal cells undergo proliferation to become luminal.

5. On the same note, Fig 5B and 5C - it would be good to quantify the percentage of cells that don't undergo basal \rightarrow luminal and/or recombination of N1ICD.

6. What cellular state do the authors speculate the cells remain in when proliferation is blocked but N1ICD is expressed?

7. Is it that one N1ICD expressing basal cell gives rise to two N1ICD expressing luminal cells that continue to proliferate and generate more HRneg progenitors? It would be great if the authors could clarify the cellular dynamics of the process.

****Other points****

Line 417 - should be lower row instead of lower raw.

There is a typo in Figure S4 A and D. Dark blue should be INT2, not INT1 as marked.

2. Significance:

Significance (Required)

Overall, the study is well conducted, well written and adds to the understanding of cellular plasticity and the progressive transition of unipotent basal cells to luminal cells through Notch signaling. As Notch signaling is an important pathway in cell fate and malignant transformation, the study will be of interest to reserachers studying the Notch pathway in normal and cancer cells. In addition the study enhances our understanding of how cellular plasticity is regulated in epithelial cells.

3. How much time do you estimate the authors will need to complete the suggested revisions:

Estimated time to Complete Revisions (Required)

(Decision Recommendation)

Between 1 and 3 months

Yes

Review Commons Refereed Preprint #RC-2024-02581, referee reports

Original comments in black, authors' replies in blue.

Reviewer #1 (Evidence, reproducibility and clarity (Required)):

In the current study, Fre and colleagues studied the mechanisms by which ectopic activation of Notch signaling in mammary basal cells promotes a **stepwise transition** from basal to luminal cell identity in adult mouse mammary glands. Also, they tested the conservation of this mechanism in other glandular epithelia such as the lacrimal gland, the salivary gland and the prostate. To uncover the molecular mechanism underlying this progressive switch, they performed single-cell transcriptomic analysis by SMART-Seq on index-sorted mutant mammary cells at different stages of lineage conversion coupled with assays in organoid cultures. They demonstrated that the **proliferation of basal cells** is required to convert them into luminal progenitors following NICD expression.

The manuscript is interesting and the data of high quality. The novelty of this manuscript relies in the use of single-cell sequencing to study the mechanisms of cell plasticity and the demonstration that this mechanism is conserved in other glandular epithelia such as lacrimal gland, salivary gland and prostate. The involvement of proliferation in this transition is a novel aspect compared to their previous study. This manuscript will be interesting for those studying stem cell and development biology, plasticity, Notch signalling and fate conversion and will appeal the broad range of readers from **EMBO J** or Developmental Cell. We provided some suggestions to further improve the manuscript.

Major comments:

1. The data suggesting that ectopic Notch activation in basal cells of **other epithelia** should be strengthened and quantified. The immunofluorescence images shown are not always clear and informative. **Quantification of cell types** should be performed as they did for mammary glands (Fig. 1 C, D, E).

We thank the reviewer for nicely summarizing our study and appreciating its broad interest.

We agree that data in the other three epithelia need to be better quantified. We will provide quantifications of the number of GFP+ cells at different stages during the transition, both in vivo and in organoids. To clarify our IF images, we will include magnification insets for the mutant cells, stained with different fate markers.

2. The number of **cells sequenced seems low**. Could the authors explain why the number of retrieved cells is low and **demonstrate that despite this low number of cells, the results obtained are reliable?**

We agree that the number of cells is limited, but our goal was to identify and thoroughly characterize cells in intermediate states during the lineage switch, which are very few in numbers at any given time. This is the primary reason for selecting the SmartSeq technology over the more broadly used 10X scRNAseq approach. Importantly, this technique allowed us to individually sort FACS-indexed single cells in 96-well plates. The advantages of this approach include the knowledge of the state of the cells when sorted (since each cell is indexed on a FACS plot where the gates for BC, LC and intermediate cells are well defined) and an improved sequencing depth, albeit for a lower number of cells than the

classical 10X droplet-based technique. In our study, a total of 744 cells were sent for sequencing and, after QC processing and filtering, 474 cells were used for analysis. This number is consistent with the number of cells sequenced in other published Smart-seq2 analyses: Wuidart et al. (2018) used a dataset retaining 193 cells after QC, and Centonze et al. (2020) retained 337 cells after QC. In terms of sequencing depth, the median in our dataset is 5991 genes detected/cell, whereas in 10x Genomics single-cell RNA sequencing experiments, the average number of genes detected per cell typically ranges from 1,000 to 3,000 genes.

To demonstrate that despite the low number of cells sequenced, the results obtained are reliable, we will provide the quality control (QC) for all analyzed cells, showing an adequate range of nFeature_RNA (number of genes detected per cell), nCount_RNA (total number of RNA molecules detected within a cell) and mitochondrial genes expression (used to exclude apoptotic cells) (Fig. R1). These metrics indicate the high depth of sequencing of our experiments.

Figure R1: Violin plot of QC metrics nFeature_RNA, nCount_RNA and percent of mitochondrial genes for the cells used for downstream analyses in our study.

To further demonstrate consistency of our sequencing results, we have provided a scatter plot (in Fig. 2C) and violin plots (in Fig. S2C), both demonstrating that published transcriptional signatures for adult BC and LC were entirely identified in our sequenced BC and LC.

3. The results concerning **proliferation linked to the intermediate clusters** should be more detailed. The authors should provide more information regarding the high proliferative group within the INT2 cluster. Which **genes relative to cell cycle** progression are upregulated in this group? It would also be important to assess in which **specific phases of the cell cycle** are all the clusters. Is the **proliferative group in S phase**?

To address this point, we will provide a heatmap listing the genes upregulated in the proliferative group (Fig. R2), many of which are associated with cell cycle progression (see arrows in Fig. R2).

Figure R2: Heatmap graph indicating the genes that are upregulated in the proliferative cluster. Purple is associated with low expression and yellow indicates high expression. Arrows indicate cell cycle related genes.

We will also provide the list of genes that we used to calculate the cell cycle score, and indicate which ones were selected as related to the GO term "Cell cycle". **Figure R3** shows that the highly proliferative group, enriched in INT1 and INT2 cells (**Fig. R3A**), presents exclusively cells in G2M and S phase (**Fig. R3B**) and shows the highest enrichment for our cell cycle score (**Fig. R3C**).

Figure R3: **A.** UMAP plot showing clustering of single sequenced cells by SmartSeq2. **B.** UMAP indicating the cell cycle and highlighting the proliferative cluster (rectangle) enriched in INT2 cells. **C.** UMAP plot showing the cell cycle phase for each sequenced cell. The boxed regions show the proliferative cluster enriched in G2M and S phase.

Figure R4: UMAP plots showing the computed G2/M (**A**) and S score (**B**), both found highly enriched in the proliferative group comprising mainly INT1 and INT2 cells (boxed area).

Finally, we have also computed G1, G2/M and S scores to provide information on the cell cycle phase of each sequenced cell. **Fig. R4** shows UMAP plots indicating that indeed cells belonging to the proliferative group are strongly enriched for the G2/M (**Fig. R4A**) and S (**Fig. R4B**) score.

4. The **difference between WT luminal cells and mutant cells** should be strengthened. The authors should provide more robust information to exemplify the difference between WT luminal cells and mutant luminal cells. Indeed, the PCA plot reported in Fig. S3B is not really showing a clear separation between the 2 groups. Moreover, only one of the feature plots (Clic6) showed a clear difference between the 2 groups. The authors should check also other genes other than the luminal markers genes to identify the underlying differences between these luminal cells.

To conduct a more detailed comparison between WT LC and mutant LC, we will provide a heatmap showing the differences between these two luminal cells and identify the genes uniquely expressed in each group and correlate them with GO and KEGG pathways.

5. The authors should perform **further analysis of the transcriptomic data** and not only compared these data with the already published dataset. By doing so, the authors might be able to propose novel molecular mechanisms driving the cell fate switch upon Notch activation.

We have performed a deep and thorough analysis of our data and unfortunately could not pinpoint any novel molecular mechanism driving the cell fate switch, but to satisfy the Reviewer's request, we propose to use the genes we identified in Fig. S5A as being differentially expressed during the transition and to perform KEGG and GO analyses along the lineage trajectory. This might point to some pathways or GO terms associated with the lineage transition that we may have missed in our previous analysis.

6. The author should provide more data regarding the difference observed between **adult and pubertal mice**. Which is the proportion of **recombined cells at T0** (just after the induction)? The difference observed in the kinetic switch between puberty and adulthood, with the given data, can be explained by differences in the recombination proportion. Also, **difference in proliferation** of the cells between puberty and adulthood should be also investigated.

To address this important point, we will provide the data showing the difference in kinetics of lineage switch between adult and pubertal mice. As suggested by the Reviewer, we will also quantify the percentage of recombined cells (GFP positive) within the total BCs at T0, thus 48h after induction, both at puberty (in 4-week-old animals) and in adult mice. To investigate potential differences in proliferative activity, we will also quantify the number of proliferative EdU+ cells at puberty and in adult mice. These experiments should clarify if the observed differences in kinetics of cell fate switch depend on different recombination efficiency or different cycling properties of pubertal vs adult BC.

7. The authors should provide more precise information regarding the **EdU quantification** presented in Fig. S6 C, D. Are the mutant basal cells more proliferative than the WT? **Quantifying the GFPpos/EdU pos basal cells** would help answer this question.

To address this point, we will provide a more comprehensive EdU quantification in a new Fig. S6C-D, by quantifying the proportion of Edu+ cells in WT and mutant BC, in control organoids (DMSO-treated) shortly upon Notch activation, when the majority of the GFP+ cells are still BC (2-3 days), and later on when the switch is complete (6 days).

8. The author should provide more information to sustain the following statement "Some mGFPpos cells instead remained in the basal compartment after mitosis, undoubtedly representing **WT BCs that only floxed the mTmG reporter** but not the more refractory N1ICD allele".

From our long-lasting experience working with these mouse lines, we know that the mTmG allele undergoes recombination at a much higher frequency than the N1ICD allele. To experimentally demonstrate this, we will provide the quantification of GFP-positive cells by FACS, 48 hours post-induction, in vivo in both SMACre/mTmG and SMACre/N1ICD mice.

Furthermore, we will perform further immunofluorescence staining for the Notch target HES1 in SMACre/N1ICD/mTmG organoids at the end of the lineage switch (after 6 days in culture), when all

GFP+ cells have switched to LC in SMACre/N1ICD animals (see Fig. 4A). At this time point, any GFP+ cell within the luminal compartment will necessarily be mutant and therefore HES1-positive, whereas the remaining GFP+ BC will represent WT cells that recombined only the fluorescent mTmG reporter but not the Notch gain-of-function allele.

9. The statement relative to **Notch activation leading to hyperplastic lesions**, reported between line 423 and 429, should be strengthened by including data in the current manuscript.

We agree that this is an important point and propose to include images of these hyperplastic lesions by H&E and IF.

Reviewer #1 (Significance (Required)):

This manuscript will interest those studying stem cell and development biology, plasticity, Notch signaling, and fate conversion. It will appeal to a **broad range of readers from EMBO J** or Developmental Cell.

Reviewer #2 (Evidence, reproducibility and clarity (Required)):

The manuscript titled "Transcriptional landscapes underlying Notch-induced lineage conversion and plasticity of mammary basal cells" by Candice Merle et al. investigates the mechanisms by which Notch signaling influences lineage conversion in mammary basal cells. The study utilizes single-cell transcriptomics and organoid cultures to explore the molecular pathways involved in this process and underscores the broader implications of these findings in other glandular epithelia. The research presents **significant insights into cellular plasticity** and the **role of Notch signaling in cancer** biology, yet some novelty is lacking with respect to previous findings regarding Notch activation in the cell state transition from basal to luminal fate. While the **research is innovative and thorough**, there are several areas that could benefit from further exploration to strengthen the findings and broaden the implications. Detailed characterization of intermediate cell states, temporal dynamics, and functional validation of key regulators are needed to enhance the robustness of the conclusions. Otherwise, this is a **very thoughtful and insightful study**.

Reviewer #2 (Significance (Required)):

1. While the study identifies intermediate cell states between the basal and luminal fates, **further characterization and functional assays** (e.g., mammosphere, proliferative potential, differentiation potential) would help to clarify the **roles and heterogeneity** of these intermediate states.

a. It would be extremely insightful to investigate **how these intermediate states contribute functionally to the lineage conversion process** and their potential implications in the transition from basal to luminal states after Notch activation.

We agree with the Reviewer; indeed, this was the original aim when we embarked on a detailed characterization of the intermediate states. However, to our disappointment, we could not associate neither of the two intermediate states to a specific and unique transcriptional signature. We found that these transitional states represent metastable temporary cell states where cells progressively lose the characteristics of BC and acquire the determinants of the luminal lineage. Because of the lack of specific and exclusive markers for INT1 and INT2, these cells are impossible to identify and sort in view of performing functional assays, as suggested by the Reviewer.

b. Have the authors attempted to perform **loss-of-function studies for candidate genes** (like Tacc3 or Racgap1) required to mediate the Notch-mediated basal to luminal commitment?

To address this important point, we will perform loss of function experiments for the genes *Tacc3* and *Racgap1* in mammary gland organoids. We will then quantify the percentage of GFP+ (mutant cells) that are able to complete the fate transition from BC to LC in the knock-out organoids.

c. Are any genes from the INT1 and INT2 clusters able to be **validated in situ**?

As explained above, we could not pinpoint any genes specifically and exclusively associated to the INT1 cluster, which shares most of its signature genes with the BC cluster, albeit their expression levels are in general reduced. However, we could identify a few genes that seemed associated to the INT2 cluster, mainly corresponding to genes involved in proliferative activity, such as the *Tacc3* and *Racgap1* genes mentioned at point 2b above. We thus propose to perform single molecule RNA FISH for these genes in sections of mammary gland from our mutant N1ICD mice to test their expression *in vivo* and *in situ*.

2. The addition of **salivary, lacrimal, and/or prostate organoids** would benefit the broader applicability of the proliferation requirement for this state transition.

According to the Reviewer's suggestion, all experiments we have performed in mammary organoids to establish the essential role of proliferation for the fate transition will be also repeated in salivary, lacrimal and prostate organoids.

3. In Figure 1 it would help to include the **quantitative data for the other organs**.

This point was also raised by Reviewer 1; we agree that data in the other three epithelia need to be more extensively quantified. We will thus provide quantifications of the number of GFP+ cells at different stages during the fate transition, both *in vivo* and in organoids.

4. For the **organoid studies** in Figure 4, it would be helpful to include **quantitation of the eGFP within basal and luminal states** over time post NICD induction.

Between day 3 and day 6 could capture more nuanced changes occurring within the transition, along with the inclusion of proliferation markers like BrdU in the analysis within these studies including the Aphidicolin and U0126 treatments.

To address this point also raised by the other two Reviewers, we will provide a more comprehensive EdU quantification in a new Fig. S6C-D, by calculating the proportion of EdU+ cells in WT and mutant BC, in control organoids (DMSO-treated) shortly upon Notch activation, when the majority of the GFP+ cells are still BC (2-3 days), and later on when the switch is complete (6 days). These quantifications will also be performed in organoids treated with aphidicolin and U0126.

5. Have the authors considered **ChIP-seq or ATAC-seq** to complement the scRNA sequencing studies that could add to the lineage trajectories identified?

This is a very good suggestion. However, such experiments can be very challenging with the low number of intermediate cells that we can isolate *in vivo*; furthermore, the fact that we could not identify a specific and exclusive transcriptional signature characterizing the intermediate state makes us fear that such difficult and rather expensive approaches might not turn out very informative and we consider this good suggestion beyond the scope of the present study.

Reviewer #3 (Evidence, reproducibility and clarity (Required)):

In the manuscript, "Transcriptional landscapes underlying Notch-induced lineage conversion and plasticity of mammary basal cells" by Merle et.al, the authors elucidate the how ectopic expression of N1ICD in basal mammary cells (and other epithelial) reprograms them to a luminal cell fate. Using single cell RNA-sequencing they find that the change is progressive from basal \diamond luminal and involves 2

intermediate stages labelled as INT1 and INT2. The authors also find that there are no unique genes expressed by these subsets, but they are similar in transcriptional space to other EMP/hybrid cells described previously. Interestingly, some of the INT2 cells show higher cell cycle score. Therefore, the authors test and conclude that proliferation of basal cells in addition to N1ICD expression is required for change to luminal identity consistent with previous studies.

Below are a few points to consider regarding the findings which in my opinion will clarify and enhance the findings of the paper.

1. Is the **INT1/INT2 state present during pubertal development in TEBs or during pregnancy**? Can the authors perform a similar **label transfer analysis using the single cell datasets generated from other studies at these developmental time points**? (OPTIONAL but will be very interesting to do)

This interesting suggestion will be taken in consideration by performing label transfer analysis using published datasets that provide the transcriptomic profile of mammary epithelial cells at puberty and pregnancy (Bach et al, 2017 and Pal et al, 2021).

2. Have the authors considered using **entropy-based tools** to understand whether the **INT1/2 state is indeed multipotent** such as **StemID** or cellular potency tools such as **CytoTRACE**? (OPTIONAL but will be interesting to do)

We have not considered these bioinformatic approaches and thank the Reviewer for the suggestion. We will implement these analysis tools to further characterize the potency of intermediate cells.

3. The authors conclude that the basal cells need to proliferate to become luminal - however in the UMAPs on Fig 2E, **most of the cells in the INT2 cluster don't have a high cell cycle score** or expression of cell cycle related genes Fig S5C? Is this due to the timing of when the cells were sorted for sequencing?

Indeed, while we show that the transition from basal to luminal cells requires proliferation, this step is transient and rapid, so we predict that the cells in mitosis within the INT2 cluster at any given time are hard to catch in transcriptomic analyses. Also, we have shown that the lineage switch is not synchronous, as some cells transition to luminal fate very rapidly upon Notch activation, and others require a much longer time. These are some of the possible explanations of why not all INT2 cells are caught during division and thus they do not all show a high cell cycle score. To better clarify this conundrum, we propose to evaluate the proportion of INT2 cells that are found enriched in a G0/G1 score that we will compute and compare it with the percentage of cells from the other clusters enriched for the same "quiescence/differentiation" score.

4. Fig. S6C not all N1ICD+ cells are EdU+ - it would be great if authors can clarify what **percent of N1ICD expressing basal cells undergo proliferation to become luminal**.

This point is related to point 7 of Reviewer 1 and comment 4 of Reviewer 2.

We agree with this concern and will provide a more comprehensive EdU quantification in a new Fig. S6C-D, by calculating the proportion of EdU+ cells in WT and mutant BC, in control organoids (DMSO-treated) shortly upon Notch activation, when the majority of the GFP+ cells are still BC (2-3 days), and later on when the switch is complete (6 days). These quantifications will also be performed in organoids treated with aphidicolin and U0126.

5. On the same note, Fig 5B and 5C - it would be good to **quantify the percentage of cells that don't undergo basal \rightarrow luminal and/or recombination of N1ICD**.

As answered to Reviewer 1, from our long-lasting experience working with these mouse lines, we know that the mTmG allele undergoes recombination at a much higher frequency than the N1ICD allele. To experimentally demonstrate this, we will provide the quantification of GFP-positive cells by FACS, 48 hours post-induction, *in vivo* in both SMACre/mTmG and SMACre/N1ICD mice.

Furthermore, we will perform further immunofluorescence staining for the Notch target HES1 in SMACre/N1ICD/mTmG organoids at the end of the lineage switch (6 days), when all GFP+ cells have switched to LC in SMACre/N1ICD animals (see Fig. 4A). At this time point, any GFP+ cell within the luminal compartment will necessarily be mutant and therefore HES1-positive, whereas the remaining GFP+ BC will represent WT cells that recombined only the fluorescent mTmG reporter but not the Notch gain-of-function allele.

6. What **cellular state** do the authors speculate the cells remain in **when proliferation is blocked but N1ICD is expressed**?

When proliferation is blocked, the mutant cells maintain expression of basal markers and remain in a basal position. Transcriptomic analysis shows that the INT1 cell cluster is more closely related to BC than to LC, leading us to speculate that the mutant cells expressing N1ICD, when proliferation is blocked in organoids, would most likely resemble cells found *in vivo* within the INT1 cluster.

7. Is it that **one N1ICD expressing basal cell gives rise to two N1ICD expressing luminal cells that continue to proliferate** and generate more HRneg progenitors? It would be great if the authors could clarify the cellular **dynamics** of the process.

To answer this interesting suggestion, we will acquire longer time lapse imaging of mutant organoids, to catch the fate of daughter cells after a division. Immunofluorescent analysis on fixed cells at the end of the live imaging should provide information on the differentiation state of the daughter cells upon division. These experiments will be performed with a multicolor reporter line (Rosa-Confetti mice) allowing us to lineage trace distinct cell division events with different colors, which should facilitate the retrospective analysis of the fate of the daughter cells resulting from each captured mitosis.

Reviewer #3 (Significance (Required)):

Overall, the **study is well conducted, well written** and adds to the understanding of cellular plasticity and the progressive transition of unipotent basal cells to luminal cells through Notch signaling. As Notch signaling is an important pathway in cell fate and malignant transformation, the study will be **of interest to researchers studying the Notch pathway in normal and cancer cells**. In addition, the study **enhances our understanding of how cellular plasticity is regulated** in epithelial cells.

Dear Silvia,

Thank you for transferring your manuscript EMBOJ-2024-119409-T | [RC-2024-02581] with Review Commons referee reports and responses to The EMBO Journal, and also for your patience with our feedback at this time of the year. I have now carefully assessed your revision plan and rebuttal to the referees at Review Commons, and in addition discussed this in the editorial team.

We acknowledge that you are planning a substantial experimental revision to complement your study and address the critique raised by the referees and consider your response to be sensible. We can thus invite you to revise your study along the lines sketched in your outline for further consideration by the EMBO Journal.

We concur that the request on additional ChIP-seq and ATAC-seq data (reviewer #2, pt.5) is well taken as such, however beyond the scope of the current work in our view.

Please feel free to contact me if you have any questions or need further input on the referee comments.

When submitting your revised manuscript, please carefully review the instructions below.

Please feel free to approach me any time should you have additional questions related to this.

Thank you for the opportunity to consider your work for publication.

I look forward to your revision.

Best regards,

Daniel

Daniel Klimmeck, PhD
Senior Editor
The EMBO Journal

Instruction for the preparation of your revised manuscript:

- 1) a .docx formatted version of the manuscript text (including legends for main figures, EV figures and tables). Please make sure that the changes are highlighted to be clearly visible.
- 2) individual production quality figure files as .eps, .tif, .jpg (one file per figure).
- 3) a .docx formatted letter INCLUDING the reviewers' reports and your detailed point-by-point response to their comments. As part of the EMBO Press transparent editorial process, the point-by-point response is part of the Review Process File (RPF), which will be published alongside your paper.
- 4) a complete author checklist, which you can download from our author guidelines ([https://wol-prod-cdn.literatumonline.com/pb-assets/embo-site/Author Checklist%20-%20EMBO%20J-1561436015657.xlsx](https://wol-prod-cdn.literatumonline.com/pb-assets/embo-site/Author%20Checklist%20-%20EMBO%20J-1561436015657.xlsx)). Please insert information in the checklist that is also reflected in the manuscript. The completed author checklist will also be part of the RPF.
- 5) Please note that all corresponding authors are required to supply an ORCID ID for their name upon submission of a revised manuscript.
- 6) It is mandatory to include a 'Data Availability' section after the Materials and Methods. Before submitting your revision, primary datasets produced in this study need to be deposited in an appropriate public database, and the accession numbers and

database listed under 'Data Availability'. Please remember to provide a reviewer password if the datasets are not yet public (see <https://www.embopress.org/page/journal/14602075/authorguide#datadeposition>).

7) Our journal encourages inclusion of *data citations in the reference list* to directly cite datasets that were re-used and obtained from public databases. Data citations in the article text are distinct from normal bibliographical citations and should directly link to the database records from which the data can be accessed. In the main text, data citations are formatted as follows: "Data ref: Smith et al, 2001" or "Data ref: NCBI Sequence Read Archive PRJNA342805, 2017". In the Reference list, data citations must be labeled with "[DATASET]". A data reference must provide the database name, accession number/identifiers and a resolvable link to the landing page from which the data can be accessed at the end of the reference. Further instructions are available at .

8) At EMBO Press we ask authors to provide source data for the main and EV figures. Our source data coordinator will contact you to discuss which figure panels we would need source data for and will also provide you with helpful tips on how to upload and organize the files.

Numerical data can be provided as individual .xls or .csv files (including a tab describing the data). For 'blots' or microscopy, uncropped images should be submitted (using a zip archive or a single pdf per main figure if multiple images need to be supplied for one panel). Additional information on source data and instruction on how to label the files are available at .

9) We replaced Supplementary Information with Expanded View (EV) Figures and Tables that are collapsible/expandable online (see examples in <https://www.embopress.org/doi/10.15252/emj.201695874>). A maximum of 5 EV Figures can be typeset. EV Figures should be cited as 'Figure EV1, Figure EV2' etc. in the text and their respective legends should be included in the main text after the legends of regular figures.

11) For data quantification: please specify the name of the statistical test used to generate error bars and P values, the number (n) of independent experiments (specify technical or biological replicates) underlying each data point and the test used to calculate p-values in each figure legend. The figure legends should contain a basic description of n, P and the test applied. Graphs must include a description of the bars and the error bars (s.d., s.e.m.).

We realize that it is difficult to revise to a specific deadline. In the interest of protecting the conceptual advance provided by the work, we recommend a revision within 3 months (23rd Jan 2025). Please discuss the revision progress ahead of this time with the editor if you require more time to complete the revisions. Use the link below to submit your revision:

Link Not Available

Rev_Com_number: RC-2024-02581
New_manu_number: EMBOJ-2024-119409-T
Corr_author: Fre
Title: Transcriptional landscapes underlying Notch-induced lineage switch and plasticity of mammary cells

EMBOJ-2024-119409-T, Response to Referee Reports

Original comments in black, authors' replies in green.

Changes are highlighted in blue in the revised manuscript.

Reviewer #1 (Evidence, reproducibility and clarity (Required)):

In the current study, Fre and colleagues studied the mechanisms by which ectopic activation of Notch signaling in mammary basal cells promotes a stepwise transition from basal to luminal cell identity in adult mouse mammary glands. Also, they tested the conservation of this mechanism in other glandular epithelia such as the lacrimal gland, the salivary gland and the prostate. To uncover the molecular mechanism underlying this progressive switch, they performed single-cell transcriptomic analysis by SMART-Seq on index-sorted mutant mammary cells at different stages of lineage conversion coupled with assays in organoid cultures. They demonstrated that the proliferation of basal cells is required to convert them into luminal progenitors following NICD expression.

The manuscript is interesting and the data of high quality. The novelty of this manuscript relies in the use of single-cell sequencing to study the mechanisms of cell plasticity and the demonstration that this mechanism is conserved in other glandular epithelia such as lacrimal gland, salivary gland and prostate. The involvement of proliferation in this transition is a novel aspect compared to their previous study.

This manuscript will be interesting for those studying stem cell and development biology, plasticity, Notch signalling and fate conversion and will appeal the broad range of readers from EMBO J or Developmental Cell. We provided some suggestions to further improve the manuscript.

We warmly thank the Referee for nicely summarizing our study and for their appreciation of the interest and novelty of our results.

Major comments:

1. The data suggesting that ectopic Notch activation in basal cells of other epithelia should be strengthened and quantified. The immunofluorescence images shown are not always clear and informative. Quantification of cell types should be performed as they did for mammary glands (Fig. 1 C, D, E).

We concur with the Referee on the necessity of strengthen and further quantify our data on other epithelia. We now provide new more informative immunofluorescence images with magnified views as insets in Fig. 1E-G for lacrimal and salivary glands and for the prostate. Additionally, as suggested, we have quantified the number of mutant GFP+ basal and luminal cells for the lacrimal and salivary glands at 3 weeks and 6 weeks post-induction. These results are now presented in the new Fig. 1D and described in the manuscript at lines 102-105. Consistent with our observations in the mammary gland (Fig. 1C), we found a progressive transition from basal to luminal lineage over 6 weeks, when most mutant cells (90 to 98% for lacrimal and salivary glands) have switched fate and become luminal.

2. The number of cells sequenced seems low. Could the authors explain why the number of retrieved cells is low and demonstrate that despite this low number of cells, the results obtained are reliable?

We agree that the number of cells sequenced is low, but our goal was to isolate and molecularly characterize cells in intermediate states of the lineage switch, which are very few in numbers at any given time. This is the primary reason for our choice of employing the SmartSeq technology over the more broadly used 10X scRNAseq approach. Importantly, this technique allowed us to individually sort FACS-indexed single cells in 96-well plates. The advantages include our knowledge of the state of the cells when sorted (since each cell is indexed on a FACS plot where the gates for BC, LC and intermediate cells are well defined) and an improved sequencing depth, albeit for a lower number of cells than the classical 10X droplet-based technique. We have sequenced a total of 744 cells and, after QC processing and filtering, 474 cells were retained for further analyses. This number is consistent with the number of cells sequenced in other published Smart-seq2 analyses: Wuidart et al. (2018) used a dataset retaining 193 cells after stringent QC, and Centonze et al. (2020) retained 337 cells after QC. In terms of sequencing depth, the median in our dataset is 5991 genes detected/cell, whereas in another 10x dataset sequenced in the lab for a different project the median was 3782 genes detected/cell, validating the improved depth and good quality of this dataset. This difference in number of genes detected and coverage has also been previously demonstrated (e.g. Wang et al., 2021).

To demonstrate that despite the low number of cells, the results we obtained are reliable, we provide here the quality control (QC) for all analysed cells, showing an adequate range of nFeature_RNA (number of genes detected per cell), nCount_RNA (total number of RNA molecules detected within a cell) and mitochondrial genes expression (used to exclude apoptotic cells) (**Fig. R1**). These metrics indicate a high depth of sequencing, and a considerable number of genes detected per cell.

To further demonstrate consistency and reliability of our sequencing results, and following the Referee's advice, we also provide a scatter plot (in Fig. 2C) and violin plots (in Fig. EV1F), and label transfer (Fig. EV3) showing that published transcriptional signatures for adult BCs and LCs coincide with and identify our sequenced BCs and LCs.

Fig. R1: Violin plots of QC metrics: nFeature_RNA, nCount_RNA and percent of mitochondrial genes for the cells used for downstream analyses in our study, demonstrating the high quality of our sequenced cells.

3. The results concerning proliferation linked to the intermediate clusters should be more detailed. The authors should provide more information regarding the high proliferative group within the INT2 cluster. Which genes relative to cell cycle progression are upregulated in this group? It would also be important to assess in which specific phases of the cell cycle are all the clusters. Is the proliferative group in S phase?

To address the Referee's comment, we have now performed a focussed analysis on the proliferative group (using CellSelector tool from Seurat package) and identified genes that are differentially expressed in the proliferative group. This analysis is now shown in Fig. EV1I, and described in the text at lines 158-162, where we highlighted several genes associated with cell cycle progression, such as *Mki67*, *Top2a*, *Cenpf* and *Aurkb*.

To provide a more comprehensive vision of the cell cycle profile of cells comprised in the proliferative group, we have now included a new panel (Fig. EV1G and described in the text at lines 154-158), which shows that these cells are exclusively in the S or G2/M phase. Additionally, we computed a G2/M and an S phase molecular score, and we show the specific enrichment of these scores in the highly proliferative cells boxed in Fig. EV1H (and described in the text at lines 155-158). Upon quantification of the proportion of cells from each cluster in G1, S or G2/M, we also provide further evidence that the highly proliferative INT2 cluster comprises fewer cells in G1 and more in G2/M phase compared to all other clusters (Fig. EV1J and described in the text at lines 160-162).

4. The difference between WT luminal cells and mutant cells should be strengthened. The authors should provide more robust information to exemplify the difference between WT luminal cells and mutant luminal cells. Indeed, the PCA plot reported in Fig. S3B is not really showing a clear separation between the 2 groups. Moreover, only one of the feature plots (Clic6) showed a clear difference between the 2 groups. The authors should check also other genes other than the luminal markers genes to identify the underlying differences between these luminal cells.

We agree and thank the Referee for this valuable suggestion.

First, to demonstrate that there is a difference in the proportion of GFP^{pos} cells between the cluster “luminal” and “pre-luminal”, we added in the new Fig. EV2B the quantification of the percentage of GFP^{pos} and GFP^{neg} cells in the two clusters, indicating a predominance of mutant cells in the pre-luminal group.

Subsequently, we generated a heatmap (in Fig. EV2E and discussed in the text at lines 176-181) showing differences in gene expression between “pre-luminal” and “luminal” cells, based on the two clusters identified in Fig. EV2B. This additional analysis highlighted several luminal genes that are less expressed in pre-luminal cells, such as *Muc1*, *Clic6*, *Lrg1*, or *Chil1*.

To broaden and reinforce our analysis and highlight differences between WT and mutant HR^{neg} luminal cells, we then selected the cells index-sorted as luminal HR^{neg} by FACS and performed DEG analysis. This query identified some genes differentially expressed in WT and mutant cells, as represented in the volcano plot that we included as Fig. EV2F and discussed in the manuscript at lines 181-184. While the unique genes specific for WT or mutant HR^{neg} LCs were not in sufficient number to perform KEGG or GO analysis, it is noteworthy that we could identify the gene *Dclk1*, associated with gastric and breast cancer (Afshar-Sterle et al., 2024; Wang et al., 2019), as uniquely and highly enriched in mutant luminal HR^{neg} cells. On the contrary, *Igfbp5*, the gene encoding the Insulin-like growth factor-binding protein 5, was strongly enriched in WT HR^{neg} cells. This gene has been reported to be causally related to apoptosis in the mammary gland (Allan et al, 2004) although the significance of its high expression in this context remains to be established.

5. The authors should perform further analysis of the transcriptomic data and not only compared these data with the already published dataset. By doing so, the authors might be able to propose novel molecular mechanisms driving the cell fate switch upon Notch activation.

To address this comment and identify differentially expressed genes during the basal to luminal transition, we examined their dynamic expression patterns of all genes which significantly vary along the basal-luminal trajectory. This extended analysis distinguished 6 different patterns of expression along the process of basal to luminal transition, as illustrated

in Fig. EV4B and Appendix Table3 and described in the manuscript at lines 306-309. Thanks to this analysis, we identified two groups enriched in basal genes (“Bas early” and “Bas late”) which include well-known basal cell markers such as *Krt15* and *Krt5*, *Sparc* or *Cxcl14*, and are linked to the GO term “extracellular matrix organization” (Fig. EV4C). Likewise, toward the end of the pseudotime, we found enrichment in the expression of genes associated with the acquisition of a luminal phenotype (“lum early” and “lum late”), including the expression of *Krt8*, *Clic6* or *Aldh1a3*.

Of interest, this new analysis highlighted two patterns of expression specifically associated with the intermediate clusters (“int early” and “int late”). The “int early” group presented enriched expression of the cell adhesion genes *Scd1* and *Neb1*, whereas genes connected to cell proliferation, including *Ccnd1* and *Mcm2*, defined the “int late” group (Appendix Table3). Thanks to the Referee’s suggestion, this improved analysis contributed to better define the sequential steps necessary for the basal-to-luminal switch: initial repression of typical basal genes, modification of cell adhesion properties, proliferation, progressive acquisition of a luminal phenotype and finally complete expression of definitive luminal markers.

6. The author should provide more data regarding the difference observed between adult and pubertal mice. Which is the proportion of recombined cells at T0 (just after the induction)? The difference observed in the kinetic switch between puberty and adulthood, with the given data, can be explained by differences in the recombination proportion. Also, difference in proliferation of the cells between puberty and adulthood should be also investigated.

We thank the Referee for this suggestion, that we have addressed by quantifying the percentage of mutant (GFP^{pos}) BCs and LCs by cytometry depending on the time of induction (at puberty or in adult mice). These new data are now shown in the new Fig. EV5A and described in the manuscript at lines 358-360. When we triggered Notch activation in pre-pubertal (3-week-old) or adult (10-week-old) mice, we found that adult mammary cells take longer to complete the transition from basal to luminal identity. While induction before puberty (at postnatal day P21) resulted in a minimum of 85% of mutant LCs within 6 weeks, in adult mice we could detect about 30% of BCs that had not yet switched fate after a 6-week chase.

We now also provide evidence indicating that the difference in kinetics of lineage conversion is related to cell proliferation. Indeed, we have quantified the proportion of Ki67-positive cells in pubertal and adult mice and found that pubertal cells proliferate more frequently than adult cells (Fig. EV5B), thus they convert to luminal fate more rapidly than the more quiescent adult mammary cells (see text at lines 360-364).

We have also followed the Referee’s suggestion to assess potential differences in recombination efficiency in pubertal vs adult mice and found no difference in the percentage of GFP-labelled cells at puberty or in adult mice upon a 48h tamoxifen pulse (at T0). These data are included in Fig. EV5C (and described in the manuscript at lines 364-366) and indicate that the delayed kinetics of adult cells do not depend on lower recombination efficiency.

7. The authors should provide more precise information regarding the EdU quantification presented in Fig. S6 C-D. Are the mutant basal cells more proliferative than the WT? Quantifying the GFP^{pos}/EdU pos basal cells would help answer this question.

We thank the Referee for allowing us to clarify this point. Fig. S6C-D (now Fig. EV5H) illustrated that treatment with Aphidicolin or U0126 indeed blocks proliferation, as expected.

The quantification was performed counting only EdU+ cells, both GFP^{pos} and GFP^{neg}. This is now better explained in the text at lines 360-366.

To properly answer the Referee's question, we have quantified the proportion of EdU positive cells in basal cells, depending on their GFP status (mutant vs WT) and we now show in Fig. EV5I that basal mutant cells indeed proliferate more than WT BCs, indicating that Notch activation promotes proliferation, obligatory for the lineage conversion (as explained in the text at lines 401-405).

8. The author should provide more information to sustain the following statement "Some mGFP^{pos} cells instead remained in the basal compartment after mitosis, undoubtedly representing WT BCs that only floxed the mTmG reporter but not the more refractory N1ICD allele.

To clarify this point, we now provide evidence showing that the N1ICD allele is recombined at a much lower frequency than the mTmG allele, as indicated in the new Fig. EV6D (described in the manuscript at lines 424-428). When we induced recombination for 48h and quantified the percentage of GFP-positive cells in SMACre^{ERT2}/N1ICD and SMACre^{ERT2}/mTmG mice, we found that the percentage of nGFP-labelled cells was around 1-2% (1.53% in the example reported in Fig. EV6D), whereas the percentage of mGFP-positive cells was about 40%.

Given the observed lack of complete overlap between mGFP^{pos} cells (from the neutral mTmG allele) and N1ICD-expressing nGFP^{pos} cells, we also quantified the percentage of mGFP-positive cells that also expressed the Notch1 direct target Hes1 in mammary organoids. While we found that around 50% of the mGFP-positive cells remained in a basal position even after 6 days of induction, these non-responsive cells invariably lacked Hes1 expression (Fig. 5D). Furthermore, we could not detect any mGFP^{pos}/Hes1^{neg} luminal cells, demonstrating that these BCs were indeed WT, hence retained unipotency and did not generate mGFP^{pos} luminal daughters (Fig. 5C, D). The time-course analysis at 3 and 6 days after 4-OHT induction clearly illustrates an increase in the proportion of mGFP^{pos}/Hes1^{pos} luminal cells, while the percentage of mGFP^{pos}/Hes1^{pos} basal cells decreases (Fig. 5D).

We believe that these new quantitative data strongly support the notion that the mGFP^{pos} cells that remain basal are indeed WT cells; we hope the Referee will agree with us on this.

9. The statement relative to Notch activation leading to hyperplastic lesions, reported between line 423 and 429, should be strengthened by including data in the current manuscript.

We concur with the Referee and to support our statement concerning the long-term effects of Notch activation, we have now included new data, showing Notch-induced hyperplastic lesions both by haematoxylin and eosin staining and by immunofluorescence for GFP, illustrating that these mammary neoplasias, that we can only observe in Notch mutant mice, are mainly composed of nGFP+ cells (Fig. EV2G-H).

Minor comments

1. Typos are found across the manuscript (missing space and missing bold font while citing figures).

We tried to correct all typos we could find, thank you for this comment.

2. The sentence reported between lines 88 and 93 is not of immediate understanding due to complicated syntax.

This sentence was modified as follows: “Importantly, the cell fate switch from BCs to LCs happens progressively over a long period of time (6 weeks). At mid-transition (3 weeks upon induction), cells featuring nuclear GFP (nGFP^{pos}), reporting Notch pathway activation, co-express basal (K14) and luminal (K8) markers (Fig. 1A, Fig. EV1A), reminiscent of embryonic multipotent MaSCs.” (lines 83-86). We hope that we made the syntax clearer.

3. Relatively to Fig. 1B: can the authors explain why the quantification on the timepoint 2w results less than 100%?

We apologize for overlooking this; we have now corrected the quantification to reach 100%.

4. Relatively to Fig. S1D: can the author explain why the gates shown in the FACS plots are different between different timepoints?

The different timepoints were performed and cell-sorted in different experiments. Each dissociation experiment can show slight variations in fluorescence intensity, therefore we re-adjust the FACS gates at each experiment, as it is common practice with flow cytometry data.

5. Relatively to Fig. 2C: authors should provide another graphical representation about the similarities between each identified cluster either to luminal or basal identity. We suggest plotting each cluster separately and drawing the diagonal in such a way to make the visualization easier.

To take into account this suggestion we now provide the requested graphical representation in new Fig. 2C. This graph provides similar results than our previous dot plot in Fig. 2C where we showed that the INT1 cluster is more similar to a basal phenotype and the INT2 is closer to HR^{neg} LCs.

6. Relatively to Fig. 4C: the graph presented is not clear. Is the graph reporting the percentage of GFP positive cells on the SMA positive cells or rather the percentage of SMA positive cells that are GFP positive?

More importantly, important quantitative data are missing in this graph. Indeed, no WT (no NOTCH activation) nor DMSO CTRL data were included in this graph.

Previous Figure 4C reported the percentage of SMA positive mutant cells, so both SMA^{pos} and GFP^{pos}. To clarify this, this graph has been changed (now in Fig. 4E), and it now represents the percentage of basal and luminal nGFP^{pos} cells in the different conditions (DMSO, Aphidicolin or washout). Also, the quantitative data on DMSO-treated control organoids for 10 days, as requested by the Referee, was included in Fig. 4E.

7. Relative to Fig. 5C: the author should provide quantitative data to corroborate their observations.

We have added quantifications of both BCs and LCs that also expressed the Notch1 direct target HES1 in mammary organoids at 3 days and 6 days in the new Fig. 5D.

Reviewer #1 (Significance (Required)):

This manuscript will interest those studying stem cell and development biology, plasticity, Notch signaling, and fate conversion. It will appeal to a broad range of readers from EMBO J or Developmental Cell.

We thank again the Referee for the kind feedback and appreciation of the interest of our study.

Reviewer #2 (Evidence, reproducibility and clarity (Required)):

The manuscript titled "Transcriptional landscapes underlying Notch-induced lineage conversion and plasticity of mammary basal cells" by Candice Merle et al. investigates the mechanisms by which Notch signaling influences lineage conversion in mammary basal cells. The study utilizes single-cell transcriptomics and organoid cultures to explore the molecular pathways involved in this process and underscores the broader implications of these findings in other glandular epithelia. The research presents significant insights into cellular plasticity and the role of Notch signaling in cancer biology, yet some novelty is lacking with respect to previous findings regarding Notch activation in the cell state transition from basal to luminal fate. While the research is innovative and thorough, there are several areas that could benefit from further exploration to strengthen the findings and broaden the implications. Detailed characterization of intermediate cell states, temporal dynamics, and functional validation of key regulators are needed to enhance the robustness of the conclusions. Otherwise, this is a very thoughtful and insightful study.

We are grateful to the Referee for appreciating the significance and appeal of our work to a broad audience.

Reviewer #2 (Significance (Required)):

1. While the study identifies intermediate cell states between the basal and luminal fates, further characterization and functional assays (e.g., mammosphere, proliferative potential, differentiation potential) would help to clarify the roles and heterogeneity of these intermediate states.

a. It would be extremely insightful to investigate how these intermediate states contribute functionally to the lineage conversion process and their potential implications in the transition from basal to luminal states after Notch activation.

We thank the Referee for their kind feedback. The Referee raises here an important point related to the molecular link between intermediate states and lineage transition; indeed, this was the original aim when we embarked on a detailed characterization of the intermediate states. However, to our disappointment, we could not associate neither of the two intermediate states to a unique transcriptional signature. We found that these transitional states represent metastable temporary cell states where cells progressively lose the characteristics of BCs and acquire determinants of the luminal lineage. Because of the lack of strongly specific and exclusive markers for INT1 and INT2, these cells are impossible to identify and sort in view of performing functional assays, as suggested by the Referee.

b. Have the authors attempted to perform loss-of-function studies for candidate genes (like *Tacc3* or *Racgap1*) required to mediate the Notch-mediated basal to luminal commitment?

To try and address this interesting suggestion, we have performed knock-down experiments targeting either *Tacc3* or *Racgap1* using adeno-associated virus (AAV) to infect mammary organoids and 24 hours after infection, we induced N1ICD expression supplementing 4-OHT to the culture medium. The organoids were then analysed by immunofluorescence after 72h when approximately 50% of the mutant cells should have transited to a luminal type.

However, the infection efficiency was extremely low (approximately 5% of the organoid cells received the viral particles), and despite several trials, we could not find any infected cell (mCherry+) that was also N1ICD mutant (i.e. expressing nGFP). We present these results in **Fig. R2** for the Referee (**Fig. R2A**), but did not incorporate it in the manuscript, as the results were inconclusive.

Furthermore, we characterized in more detail the expression of these two genes and found that it was restricted to the cells belonging to the proliferative group (**Fig. R2B**). Indeed, these two candidate genes play an important role in cell proliferation: *Racgap1* is involved in cytokinesis and *Tacc3* is associated with mitotic spindle stability and assembly. We therefore speculate that, regardless of the organoid infection efficiency, the expression of these genes may be essential for dividing cells and their knock-down could therefore result in the specific lethality of mutant cells, possibly explaining why we could not detect any mutant infected cell.

Figure R2. A. Representative images of organoids following infection with AAV (mCherry+) targeting *TACC3* knockdown and fixed 72 hours after activation of NOTCH1. **B.** Expression of *Racgap1* and *Tacc3*. The boxed region represents the proliferative cluster previously described.

c. Are any genes from the INT1 and INT2 clusters able to be validated in situ?

As explained above, we could not pinpoint any genes specifically and exclusively associated to the INT1 cluster, which shares many signature genes with the BAS cluster. However, we could identify a few genes that seemed associated to the INT2 cluster, mainly corresponding to genes involved in proliferative activity, such as the *Tacc3* and *Racgap1* genes mentioned at point 2b above. As shown in Figure R2B, the expression of these genes in our RNA sequencing dataset is restricted to the proliferative group. To address the referee's request, we performed single-molecule RNA FISH for *Tacc3* and observed very low expression, limited to a few mammary epithelial cells. Given this minimal expression, we could not detect any difference between mutant and WT cells (**Fig. R3**).

Importantly, we believe that we cannot identify genes specific and unique for INT1 or INT2 because the progressive transition from the basal to the luminal state may not require the acquisition of a new unique transcriptional signature. Based on our accumulated results, the only necessary step for the lineage switch is the proliferation of mutant cells, hence we detect a transient upregulation of proliferative genes such as *Tacc3* and *Racgap1* in the intermediate cluster INT2, the most highly proliferative.

Figure R3. Representative images of smRNA FISH targeting *Tacc3* on mammary gland section. EpCAM stained MECs in white and N1ICD is in green. Nuclei are stained with DAPI (blue). Scale bar represents 50 μ m. Filled arrowhead indicates mutant cell. Empty arrowhead indicates cells expressing *Tacc3*.

2. The addition of salivary, lacrimal, and/or prostate organoids would benefit the broader applicability of the proliferation requirement for this state transition.

We thank the Referee for their suggestion. We have now performed new experiments establishing the essential role of proliferation for fate transition in salivary, lacrimal and prostate organoids, in addition to the mammary gland, and confirmed that proliferation is essential for lineage switch in all these tissues. These results are now presented in Fig. 4 and Fig. EV6A-C and discussed at lines 404-405 of the revised text.

3. In Figure 1 it would help to include the quantitative data for the other organs.

We concur with this comment, which was also requested by Referee #1 (please refer to our answer to point 1 of Referee #1). We have now quantified the number of mutant BCs and LCs cells in the salivary and lacrimal glands at 3 weeks and 6 weeks post-induction. These results are now presented in the new Fig. 1D and described in the manuscript at lines 102-105. Consistent with our observations in the mammary gland, we found a progressive transition from basal to luminal lineage over 6 weeks, when most mutant cells (90 to 98% for lacrimal and salivary glands) have switched fate and become luminal.

4. For the organoid studies in Figure 4, it would be helpful to include quantitation of the eGFP within basal and luminal states over time post NICD induction.

Between day 3 and day 6 could capture more nuanced changes occurring within the transition, along with the inclusion of proliferation markers like BrdU in the analysis within these studies including the Aphidicolin and U0126 treatments.

To address this point, which was also raised by the other two Referees, we have now included a new figure panel in Fig. EV5E presenting the quantification of basal and luminal GFP^{pos} cells

over time in control mammary organoids and at different times after N1ICD expression. We also included quantifications of the transition in organoids from the other tissues in Fig. 4B, E.

5. Have the authors considered CHIP-seq or ATAC-seq to complement the scRNA sequencing studies that could add to the lineage trajectories identified?

This is an excellent point and an insightful suggestion. However, the proposed approaches would be very challenging with the limited number of intermediate cells that we can isolate at any given time *in vivo*, insufficient for further single cell analyses. Furthermore, the fact that we could not identify a specific and exclusive transcriptional signature characterizing the intermediate states (see our answer to point 1) makes us fear that such difficult and rather expensive experiments might not turn out very informative and we thus consider this very good suggestion beyond the scope of the present study.

Reviewer #3 (Evidence, reproducibility and clarity (Required)):

In the manuscript, "Transcriptional landscapes underlying Notch-induced lineage conversion and plasticity of mammary basal cells" by Merle et.al, the authors elucidate the how ectopic expression of N1ICD in basal mammary cells (and other epithelial) reprograms them to a luminal cell fate. Using single cell RNA-sequencing they find that the change is progressive from basal \diamond luminal and involves 2 intermediate stages labelled as INT1 and INT2. The authors also find that there are no unique genes expressed by these subsets, but they are similar in transcriptional space to other EMP/hybrid cells described previously. Interestingly, some of the INT2 cells show higher cell cycle score. Therefore, the authors test and conclude that proliferation of basal cells in addition to N1ICD expression is required for change to luminal identity consistent with previous studies.

Below are a few points to consider regarding the findings which in my opinion will clarify and enhance the findings of the paper.

1. Is the INT1/INT2 state present during pubertal development in TEBs or during pregnancy? Can the authors perform a similar label transfer analysis using the single cell datasets generated from other studies at these developmental time points? (OPTIONAL but will be very interesting to do)

We thank the Referee for their appreciation of our study and for this suggestion. Accordingly, we have now used the same label transfer analysis to assess if the intermediate transcriptional states that we have identified can be found in physiological glands at specific times of major tissue remodelling, such as in puberty or at pregnancy and included it in Fig. EV3F-G.

Using our dataset as a reference for comparisons with cell populations in pubertal TEBs and ducts (Pal et al., 2021) or during pregnancy (Bach et al., 2017), we found that basal cells in TEBs (usually referred to as cap cells) are more similar to our INT1 cluster than to our BAS cluster. Likewise, the HR^{neg} TEBs cells correspond to our INT2 cluster instead of pairing with our HR^{neg} cluster. These results could be interpreted as suggesting that the intermediate cell states we have identified in this study resemble the less differentiated pubertal TEBs cells which have not yet acquired the full repertoire of luminal or basal definitive markers characterizing mature cells. This finding is corroborated by the fact that pubertal ductal BCs indeed cluster with our BAS cells. However, the HR^{neg} ductal cells in puberty again correlated

to our INT2 cluster rather than to HR^{neg} cells. This might indicate that ductal basal cells are more mature than the cap cells in TEBs, thus they correspond to adult basal cells, whereas HR^{neg} pubertal cells, even in ducts, are less mature than adult HR^{neg} cells.

In addition, we compared our dataset with sequencing results obtained in pregnant mice and found that basal Bsl-G (Basal cells at gestation) were similar to our BAS cells, whereas Avd (alveolar differentiated cells) corresponded in part to our HR^{neg} and some to our INT2. These findings lead us to suggest that the Avd cluster comprises both differentiating alveolar cells (the fraction pairing with our INT2 cells) and more differentiated alveolar cells that correspond to our HR^{neg} luminal cluster.

Overall, this additional analysis indicates that our intermediate clusters represent less differentiated cell states than adult BCs or LCs, resembling pubertal cells found in TEBs. These results are now presented in the text at lines 242-261.

2. Have the authors considered using entropy-based tools to understand whether the INT1/2 state is indeed multipotent such as StemID or cellular potency tools such as CytoTRACE? (OPTIONAL but will be interesting to do).

Figure R4. A-B: CytoTRACE analysis. (A) UMAP plot representing the potency score of each sequenced cell. (B) Bar plot representing the potency score per cluster. **C-D:** StemID analysis represented on the UMAP and as a bar plot with the potency/StemID score represented as the number of links * the delta-entropy per cluster.

We had not considered this and thank the Referee for their suggestion. Following the Referee's advice, we have now used both the CytoTRACE and StemID analysis tools to try to reveal the potency of intermediate cells (Fig. R4). The CytoTRACE tool classified our basal and INT1 cells as oligopotent. The predicted stem cell potency decreased along the lineage transition, with INT2 cells in between oligopotent and unipotent and luminal cells as unipotent (Fig. R4A-B). The StemID tool instead assigned the highest potency to the INT1 and INT2 cluster, but HR^{pos} luminal cells also featured a higher potency score than basal or luminal HR^{neg} cells (Fig. R4C-D). Given that our intermediate cells do not re-acquire multipotency, as

shown in Fig. EV3B-C by label transfer comparison with embryonic datasets and considering the current evidence for unipotency of adult mammary cell types, these results are contradictory, possibly indicating that they are not best suited to explore tissue-specific stem and progenitor cells, but rather pluripotent ES or iPS cells. Therefore, we prefer not to include this ambiguous analysis in our manuscript, unless the Referee deems it necessary.

3. The authors conclude that the basal cells need to proliferate to become luminal - however in the UMAPs on Fig 2E, most of the cells in the INT2 cluster don't have a high cell cycle score or expression of cell cycle related genes Fig S5C? Is this due to the timing of when the cells were sorted for sequencing?

We are happy to clarify this point, thank you. To address the Referee's comment, also raised at point 3 of Referee #1, we have now extended our analysis of the cell cycle phase of each sequenced cell and performed re-clustering of our dataset to identify genes that are differentially expressed in the proliferative group. This new examination is now shown in Fig. EV1I, and described in the text at lines 158-159, where we highlighted several genes associated with cell cycle progression, such as *Mki67*, *Top2a*, and *Aurkb*. To provide a more comprehensive vision of the cell cycle profile of cells comprised in the proliferative group, we have now included a new panel (Fig. EV1G and described in the text at lines 154-158), which shows that these cells are exclusively in the S or G2/M phase. Additionally, we computed a G2/M and an S phase molecular score and show specific enrichment of these scores in the highly proliferative cells boxed in Fig. EV1H (and described in the text lines 155-158). Upon quantification of the proportion of cells from each cluster in G1, S or G2/M, we also provide further evidence that the highly proliferative INT2 cluster comprises fewer cells in G1 phase and more in G2/M phase compared to all other clusters (Fig. EV1J and described in the text lines 160-162).

In Fig. 2E, we show that the highest cell cycle score is found in cells belonging to the INT2 cluster. Indeed, while we have discovered that the transition from basal to luminal cells requires proliferation, we believe that this step is transient and rapid, so we agree with the Referee that cells in mitosis within the INT2 cluster (in blue in new Fig. EV1J) at any given time are hard to catch in transcriptomic analyses. Also, our results clearly show that the lineage switch is not synchronous, as some cells transition to luminal fate very rapidly upon Notch activation, while others require a much longer time.

These are some of the possible explanations of why not all INT2 cells are caught during division and thus they do not all show a high cell cycle score at the time the cells were sorted for sequencing, as correctly pointed out by the Referee.

4. Fig. S6C not all N1ICD+ cells are EdU+ - it would be great if authors can clarify what percent of N1ICD expressing basal cells undergo proliferation to become luminal.

This point is related to point 7 of Referee #1 and comment 4 of Referee #2.

To address this concern, we now provide quantifications of EdU+ and EdU- basal cells separated between WT and mutant cells (Fig. EV5I). Thank to this improved analysis, we show that mutant basal cells proliferate more frequently than WT basal cells. Our results also demonstrate that eventually every N1ICD-expressing basal cell will become luminal. However, if they all need to divide to complete the lineage switch is difficult to say. Based on our experiments in organoids treated with Aphidicolin or U0126, we could establish that in these

conditions up to 20 to 25% of the cells are still able to become luminal when proliferation is blocked (Fig. 4B), but other caveats could explain this phenotype, including a lack of complete inhibition of cell division in these experiments, allowing escapee cells to lineage convert.

5. On the same note, Fig 5B and 5C - it would be good to quantify the percentage of cells that don't undergo basal \rightarrow luminal and/or recombination of N1ICD.

As answered also to point 8 of Referee #1, from our long-lasting experience working with these mouse lines, we know that the mTmG allele undergoes recombination at a much higher frequency than the more refractory N1ICD flox allele. To demonstrate this, we now provide *in vivo* quantification of GFP-positive BCs by FACS, 48 hours post-induction, in both SMACre/mTmG and SMACre/N1ICD mice. These experiments show that the percentage of nGFP-labeled cells (N1ICD+) is around 1-2% of the total BCs (Fig. EV6D, and in the text lines 424-428), whereas the percentage of mGFP-positive control cells was much higher, representing about 40% of the total BCs in SMACre^{ERT2}/mTmG mice.

Furthermore, and more compellingly, the new Fig. 5D presents the quantification of the percentage of mGFP-positive cells that also expressed the Notch1 direct target HES1 in mammary organoids. The time-course analysis at 3 and 6 days after 4-OHT induction clearly illustrates an increase in the proportion of mGFP^{pos}/Hes1^{pos} luminal cells, while the percentage of mGFP^{pos}/Hes1^{pos} basal cells decreases (Fig. 5D).

6 days after 4-OHT induction, we detected 40% of the cells that were WT basal cells, since they did not feature HES1 expression. Nearly 50% of the cells were both mGFP-positive and HES1-positive and were found in a luminal position, indicating they were N1ICD mutants. Approximately 5% of the cells were mGFP-positive and HES1-positive, but they were still found within the basal compartment, suggesting they were mutant cells that had not yet undergone the transition from basal to luminal. Importantly, no luminal mGFP-positive cell and HES1^{neg} was detected, suggesting that indeed all mGFP+ cells that were found in the luminal layer were derived from N1ICD-expressing basal cells. These experiments are now presented in the text at lines 429-438.

We hope that these additional quantifications have clarified this issue and convinced the Referee.

6. What cellular state do the authors speculate the cells remain in when proliferation is blocked but N1ICD is expressed?

When proliferation is blocked, we found that mutant cells maintain expression of basal markers and remain in a basal position. Since our transcriptomic analysis clearly indicates that the INT1 cell cluster is more closely related to BCs than to LCs, we believe that the mutant cells expressing N1ICD, when proliferation is blocked in organoids, would most likely resemble cells found *in vivo* within the BAS or the INT1 cluster, but these are pure speculations.

7. Is it that one N1ICD expressing basal cell gives rise to two N1ICD expressing luminal cells that continue to proliferate and generate more HRneg progenitors? It would be great if the authors could clarify the cellular dynamics of the process.

This is a difficult point to address *in vivo*. The dynamics of mutant cells could be followed only in organoids by time-lapse microscopy, and we observed that after cell division the two daughter cells would often move to a more internal position within organoids (Fig. 5A).

However, the current lack of live reporters for the luminal and basal identity precludes the formal conclusion that the daughters of a mutant basal cell have already become luminal. Our interpretation is that there would be intermediate states within the first few cell divisions before the final acquisition of a mature luminal fate. Then, mutant HR^{neg} cells can considerably proliferate to generate more mutant HR^{neg} cells as shown in Fig. 4A illustrating an organoid almost exclusively composed of mutant cells, often found in our culture wells, albeit the low number of recombined cells at T0.

Other points

Line 417 - should be lower row instead of lower raw.

This has been corrected, thank you.

There is a typo in Figure S4 A and D. Dark blue should be INT2, not INT1 as marked.

Thank you for spotting this, it has been fixed.

Reviewer #3 (Significance (Required)):

Overall, the study is well conducted, well written and adds to the understanding of cellular plasticity and the progressive transition of unipotent basal cells to luminal cells through Notch signaling. As Notch signaling is an important pathway in cell fate and malignant transformation, the study will be of interest to researchers studying the Notch pathway in normal and cancer cells. In addition, the study enhances our understanding of how cellular plasticity is regulated in epithelial cells.

We sincerely thank the Referee for their appreciation of the quality of our results and the interest and novelty of our study to a broad audience of researchers.

Dear Silvia,

Thank you for submitting your revised manuscript (EMBOJ-2024-119409R) to The EMBO Journal, as well for your patience with our response. Your amended study was sent back to the three referees for their scientific reassessment, and we have received detailed re-reports from all of them, which I enclose below. As you will see, the experts state that the work has been substantially enhanced by the revisions and they are now broadly in favour of publication.

Thus, we are pleased to inform you that your manuscript has been accepted in principle for publication in The EMBO Journal.

Please consider the remaining minor point by referee #1 carefully by revising the data or alternatively removing the respective analyses and adjusting claims accordingly.

Also, we now need you to take care of a number of issues related to formatting and data presentation as detailed below, which should be addressed at re-submission.

Please contact me at any time if you have additional questions related to below points.

As you might remember from previous exchange, every paper at the EMBO Journal now includes a 'Synopsis', displayed on the html and freely accessible to all readers. The synopsis includes a 'model' figure as well as 2-5 one-short-sentence bullet points that summarize the article. I would appreciate if you could provide this figure and the bullet points.

Thank you for giving us the chance to consider your manuscript for The EMBO Journal. I look forward to your final revision.

Again, please contact me at any time if you need any help or have further questions.

Best regards,

Daniel

>> Figures in separate files: EV figures should be uploaded as individual, high-resolution figure files.

>>Appendix: please add page numbers to the table of contents and correct the nomenclature in the legends to "Appendix Table S1" etc. .

>> Data availability section: Please remove the referee token and make sure the datasets are publicly accessible.

>> Source data: as to our journal policies we kindly request additional source data for Figure EV5.

>> Remove the Reagents and Tools table from the manuscript and provide as a separate file using the existing template in the Guide For Authors, listing key reagents, experimental models, software and relevant equipment.

>> Consider additional changes and comments from our production team as indicated below:

- Figure legends:

1. Please note that the legend for figure 4 is not provided in the sequential manner (legend for figure 4C, D is provided before

legend of figure 4B). This needs to be rectified.

2. Please note that the exact p values are not provided in the legend of figure EV5 H.

3. Please indicate the statistical test used for data analysis in the legend of figure EV4 C.

4. Please indicate what */ **/ ***/ **** represents; if this represents p value(s), please indicate the statistical test used and where appropriate, and the exact p value in the legend(s) of figure(s) EV5 B.

5. Please note that information related to n is missing in the legends of figures 1C, D; 4B, E; EV1 F, EV2 F, EV5 A, E, H, I.

6. Please note that the error bars are not defined in the legends of figures 4B, EV5 A, H, I.

7. Please note that the scale bar needs to be defined for figures EV1 A, EV5 F.

Referee #1:

In the revised manuscript, Silvia Fre and colleagues confirmed that the ectopic activation of Notch signalling in mammary basal cells promotes a stepwise transition from basal to luminal cell identity in adult mouse mammary gland. Also, they tested the conservation of this mechanism in other glandular epithelia such as lacrimal gland, the salivary gland and the prostate. To uncover the molecular mechanism underlying this progressive switch, they performed single cell transcriptomic analysis by SMART-Seq on index-sorted mutant mammary cells at different stages of lineage conversion coupled with assays in organoid cultures. They demonstrated that proliferation of basal mutant cells is indispensable to convert basal cells into luminal progenitors.

The author provided new strong data to support their claims. The authors addressed accurately to each reviewers' questions improving the strength of their findings.

Major comments

1. The data suggesting that ectopic Notch activation in basal cells of other epithelia should be strengthened and quantified. The immunofluorescence images shown are not always clear and informative. Quantification of cell types should be performed as they did for mammary glands (Fig. 1 C, D, E).

The authors provided quantifications and better staining. Although quantification in the Fig.1D, which is referring to the quantification of LC and BC in lacrimal gland, is not very convincing. Indeed, The SD (or SEM) is too big. Also, how can the authors explain the total absence of "intermediate" cells in LG and SG?

2. The number of cells sequenced seems very low. Could the authors explain and justify why the number of retrieved cells is so low and demonstrate that the results obtained are reliable enough?

The author replied extensively to the request and proved the robustness of their methods.

3. The results concerning proliferation linked to the intermediate clusters should be more detailed. The authors should provide more information regarding the high proliferative group within the INT2 cluster. Which genes relative to cell cycle progression are up regulated in this group? It would be also important to assess in which specific phase of the cell cycle are all the clusters. Is the proliferative group in S phase?

The author answered to these questions with a computational approach demonstrating that indeed intermediate cells have a higher proliferation genes expression. However, some further validations (i.e. in situ or BrDu FACS analysis) would have strengthen this observation.

4. The difference between WT luminal cells and mutant cells should be strengthened. The authors should provide more robust information to exemplify the difference between WT luminal cells and mutant luminal cells. Indeed, the PCA plot reported in Fig. S3B is not really showing a clear separation between the 2 groups. Moreover, only one of the feature plots (Clic6) showed a clear difference between the 2 groups. The authors should check also other genes other than the luminal markers genes to identify the underlying differences between these luminal cells.

The authors performed some comparative analysis on the DEG between the WT LCER- and the mutant luminal cells. They did not find any difference in the classical markers of ER- but they found that these 2 populations have different expression of Dcl1 and Igfbp5. Hence, the authors demonstrated that the WT LCER- and the mutant luminal cells are indeed different.

5. The authors should dig deeper into the analysis of the transcriptomic data and not stop to the mere comparison with the already published dataset. Doing this, the overall message of the manuscript is not novel, in fact, the authors did not propose

any possible molecular mechanisms driving the cell fate switch upon Notch activation is reported in the current study. The SCENIC analysis reported in the study highlighted the progressive loss and acquisition respectively of basal and luminal markers during the cell fate transition, which is something expected and not really novel. The authors provided further data assessing the gain and loss of basal and luminal phenotype. No functional data have been added to improve the manuscript novelty.

6. The authors should provide more data regarding the difference observed between adult and pubertal mice. Which is the proportion of recombined cells at T0 (just after the induction)? The difference observed in the kinetic switch between puberty and adulthood, with the given data, can be explained by differences in the recombination proportion. Also, difference in proliferation of the cells between puberty and adulthood should be also investigated.

The authors addressed this point providing data on the induction efficiency at T0, differences in the proliferation between puberty and adulthood.

7. The authors should provide more precise information regarding the EdU quantification presented in Fig. S6 C-D. Are the mutant basal cells more proliferative than the WT? Quantifying the GFPpos/EdU pos basal cells would help answer this question.

The authors answered this point extensively.

8. The author should provide more information to sustain the following statement "Some mGFPpos cells instead remained in the basal compartment after mitosis, undoubtedly representing WT BCs that only floxed the mTmG reporter but not the more refractory N1ICD allele.

The authors provided new quantitative data supporting that some mGFP positive cells are refractory to N1ICD recombination and therefore they are GFP positive but WT.

9. The statement relative to Notch activation leading to hyperplastic lesions, reported between line 423 and 429, should be strengthened by including data in the current manuscript

The authors addressed this point providing both IHC and IF showing extensive hyperplasia.

Minor comments

1. Typos are found across the manuscript (missing space and missing bold font while citing figures).

Ok.

2. The sentence reported between lines 88 and 93 is not of immediate understanding due to complicated syntax.

Ok.

3. Relatively to Fig. 1B: can the authors explain why the quantification on the timepoint 2w results less than 100%?

Ok.

4. Relatively to Fig. S1D: can the author explain why the gates shown in the FACS plots are different between different timepoints?

Ok.

5. Relatively to Fig. 2C: authors should provide another graphical representation about the similarities between each identified cluster either to luminal or basal identity. We suggest plotting each cluster separately and drawing the diagonal in such a way to make the visualization easier.

Ok.

6. Relatively to Fig. 4C: the graph presented is not clear. Is the graph reporting the percentage of GFP positive cells on the SMA positive cells or rather the percentage of SMA positive cells that are GFP positive? More importantly, important quantitative data are missing in this graph. Indeed, no WT (no NOTCH activation) nor DMSO CTRL data were included in this graph.

Ok.

7. Relative to Fig. 5C: the author should provide quantitative data to corroborate their observations.

Ok.

Referee #2:

The authors have nicely addressed the main concerns of all three reviewers and the manuscript in our opinion is now suitable for publication.

Referee #3:

The manuscript has significantly improved since its initial submission, and my previous comments have been adequately addressed. I have no further comments and fully support its publication in EMBO.

Editorial requests

Please consider the remaining minor point by referee #1 carefully by revising the data or alternatively removing the respective analyses and adjusting claims accordingly.

Referee #1:

1. The data suggesting that ectopic Notch activation in basal cells of other epithelia should be strengthened and quantified. The immunofluorescence images shown are not always clear and informative. Quantification of cell types should be performed as they did for mammary glands (Fig. 1 C, D, E).

The authors provided quantifications and better staining. Although quantification in the Fig.1D, which is referring to the quantification of LC and BC in lacrimal gland, is not very convincing. Indeed, The SD (or SEM) is too big.

We agree that the SEM is large for the salivary glands at 3 weeks from induction, but this reflects the high heterogeneity at this mid-transition timepoint, when some cells have already switched fate and others are still basal, so we would rather not change the graph or remove outliers, since it represents the actual data counted.

Also, how can the authors explain the total absence of "intermediate" cells in LG and SG?

Yes, we can. The quantifications in Fig. 1D were performed on immunofluorescence slides, manually counting basal or luminal cells. The quantification of intermediate cells could only be achieved on flow cytometry data. We have now clarified the method used for these quantifications in the legend of Fig1D.

Also, we now need you to take care of a number of issues related to formatting and data presentation as detailed below, which should be addressed at re-submission.

As you might remember from previous exchange, every paper at the EMBO Journal now includes a 'Synopsis', displayed on the html and freely accessible to all readers. The synopsis includes a 'model' figure as well as 2-5 one-short-sentence bullet points that summarize the article. I would appreciate if you could provide this figure and the bullet points.

We have included a "Synopsis" with a "model" schematic figure summarizing the major findings of the article.

Dear Silvia, dear Candice,

Thank you for submitting the revised version of your manuscript. I have now evaluated your amended manuscript and concluded that the remaining minor concerns have been sufficiently addressed.

I am thus pleased to inform you that your manuscript has been accepted for publication in the EMBO Journal.

Related, I would like to hereby ask your consent on keeping the referee response figures included in this file.

On a different note, I would like to alert you that EMBO Press offers a format for a video-synopsis of work published with us, which essentially is a short, author-generated film explaining the core findings in hand drawings, and, as we believe, can be very useful to increase visibility of the work. Please see the following link for representative examples and their integration into the article web page:

<https://www.embopress.org/doi/full/10.15252/emj.2019103932>

Best regards,

Daniel

Daniel Klimmeck, PhD
Senior Editor
The EMBO Journal
EMBO
Postfach 1022-40
Meyerhofstrasse 1
D-69117 Heidelberg
contact@embojournal.org
